# Eye blinks synchronize with musical beats during music listening

Yiyang Wu[1,2,3], Xiangbin Teng[4,5]*, Yi Du[1,2,6]*

1 State Key Laboratory of Cognitive Science and Mental Health, Institute of Psychology, Chinese Academy of Sciences, Beijing, China, 2 Department of Psychology, University of Chinese Academy of Sciences, Beijing, China, 3 Department of Psychiatry, The Affiliated Brain Hospital of Nanjing Medical University, Nanjing, China, 4 Department of Psychology, The Chinese University of Hong Kong, Shatin, New Territories, Hong Kong SAR, China, 5 Brain and Mind Institute, The Chinese University of Hong Kong, Shatin, New Territories, Hong Kong SAR, China, 6 Chinese Institute for Brain Research, Beijing, China

* Xiangbinteng@cuhk.edu.hk (XT); duyi@psych.ac.cn (YD)

## Abstract

Auditory-motor synchronization, the alignment of body movements with rhythmic patterns in music, is a universal human behavior, yet its full scope remains incompletely understood. Through four experiments with 123 young nonmusicians, integrating eye-tracking, neurophysiological recordings, white matter structural imaging, and behavioral analysis, we reveal a previously unrecognized form of synchronization: spontaneous eye blinks synchronize with musical beats. Blinks robustly synchronized with beats across a range of tempi and independently of melodic cues. Electroencephalogram recordings revealed a dynamic correspondence between blink timing and neural beat tracking. Blink synchronization performance was linked to white matter microstructure variation in the left superior longitudinal fasciculus, a key sensorimotor pathway. Additionally, the strength of blink synchronization reflected the modulation of dynamic auditory attention. These findings establish blink synchronization as a novel behavioral paradigm, expanding the auditory-motor synchronization repertoire and highlighting the intricate interplay between music rhythms and oculomotor activity. This discovery underscores a cross-modal active sensing mechanism, offering new insights into embodied music perception, rhythm processing, and their potential clinical applications.

## Introduction

Auditory-motor synchronization—the coordination of body movements with auditory rhythms, such as tapping our feet or fingers to the musical beat—represents a universal human behavior. This phenomenon, however, is relatively rare among species closely related to humans, suggesting the evolution of specialized auditory-motor neural circuitry within the human brain [1,2]. Given that deeper understanding of

**Data availability statement:** Music stimuli and preprocessed data that support the findings of this work are available in the Open Science Framework repository (https://osf.io/xhfn4/).

**Funding:** This work was supported by the Science and Technology Innovation 2030-Brain Science and Brain-inspired Artificial Intelligence Major Project (STI 2030—Major Project No. 2021ZD0201500) (https://www.most.gov.cn/index) to Y.D. and the Research Grants Council of the Hong Kong Special Administrative Region, China (Early Career Scheme; Project No. 24618124) (https://www.ugc.edu.hk/eng/rgc/) to X.T. The funders had no role in study design, data collection and analysis, decision to publish, or preparation of the manuscript.

**Competing interests:** The authors have declared that no competing interests exist.

**Abbreviations:** AF, arcuate fasciculus; ASAP, action simulation for auditory prediction; DAT, dynamic attending theory; dSLF, dorsal SLF; DTI, diffusion tensor imaging; DWI, diffusion-weighted imaging; ECG, electrocardiography; EEG, electroencephalogram; EOG, electrooculography; FA, fractional anisotropy; FOD, fiber orientation distributions; FEF, frontal eye field; FFT, Fast Fourier Transform; Gold-MSI, Goldsmiths Musical Sophistication Index; ICA, independent component analysis; LI, lateralization indices; MCCA, multiway canonical correlation analysis; MET, musical ear test; MI, mutual information; NDI, neurite density index; NODDI, neurite orientation dispersion and density imaging; ODI, orientation dispersion index; PEF, parietal eye field; pSLF, posterior SLF; RMS, root mean square; ROIs, regions of interest; SLF, superior longitudinal fasciculus; TRF, temporal response function; vSLF, ventral SLF.

behaviors often heralds deeper insights into neural systems underlying them [3], studying auditory-motor synchronization not only advances our knowledge of its distinct neural circuitry but may also offer valuable perspectives on brain disorders. Rhythm and timing deficits, for instance, are key vulnerabilities in neurodevelopmental conditions like autism spectrum disorder and developmental language disorder [4]. Moreover, auditory-motor synchronization has already proven effective in therapeutic and educational contexts for addressing motor or language disorders [5,6]. Despite extensive research, the full spectrum of auditory-motor synchronizing behaviors remains incompletely characterized. In this study, we unveil a previously unrecognized form of synchronization: spontaneous eye blinks that synchronize with musical beats. This discovery highlights a novel functional and neural linkage between cortical auditory processing and oculomotor mechanisms.

Auditory-motor synchronization is typically associated with overt actions such as finger tapping, clapping, dancing, or whispering, where body movements align with rhythmic auditory stimuli [7,8]. These voluntary movements are thought to engage both cortical and subcortical pathways, involving interactions between the auditory cortex, motor and premotor cortex, basal ganglia, and cerebellum [9–13]. According to the active sensing [14,15] and predictive coding [16,17] models in auditory contexts, and the "action simulation for auditory prediction" (ASAP) hypothesis [18] for musical beat perception in particular, motor recruitment helps refine temporal predictions of auditory patterns, optimize attention allocation, and improve auditory perception [19–21].

However, synchronized behaviors may extend beyond overt voluntary movements to include more subtle, involuntary motor actions. Music has profound effects on emotions [22], often engaging the dopamine system [23], modulating cognitive processes [24], and influencing peripheral autonomic systems [25]. These effects raise an intriguing question: could music rhythmically entrain spontaneous behaviors such as eye blinks—small but frequent motor actions with established links to dopaminergic function [26], attention [27], and cognitive effort [28]. Research on sequence processing has demonstrated that eye blinks can track temporal regularities of both linguistic structures in speech and nonlinguistic structures in isochronous tone and visual sequences [29], reflecting temporal attention. With regard to rhythm perception, while prior research has demonstrated that pupil dynamics can correlate with rhythmic structures [30], and that saccadic eye movements can synchronize with auditory rhythms [31], the relationship between eye blinks and musical rhythms remains unexplored. Eye blinks have also been studied in relation to emotion [32], attention [33], and subjective states [34] during music listening, but their potential synchronization with musical rhythms presents a novel avenue for investigation.

In this study, we investigate the synchronization of eye blinks with musical beats, a fundamental musical rhythm. Through a combination of behavioral, neurophysiological, and neuroimaging analyses, we provide robust evidence for this phenomenon, uncover its neural and structural substrates, and explore its functional role. Participants listened to Western classical music with highly regular beat patterns while their eye blinks were monitored, without any specific instructions regarding blinking.

Electroencephalogram (EEG) responses were recorded to examine how musical structures are encoded in the brain and their correlation with blink activity. Furthermore, we analyzed the microstructural properties of white matter tracts connecting the frontal, parietal, and auditory regions to identify structural differences linked to individual variation in auditory-oculomotor synchronization. Finally, we investigated the functional significance of this phenomenon, hypothesizing that stronger blink synchronization would correlate with better performance for events that aligned with the anticipated beat, consistent with the dynamic attending theory (DAT) [35–37]. Overall, this research reveals a new form of auditory-motor synchronization—blink synchronization—in music listening, and offers a mechanistic explanation through integrated behavioral, neural, and structural evidence. This work broadens our understanding of the intricate relationship between auditory rhythms and oculomotor behavior, while also laying the groundwork for future studies on its potential clinical applications, especially as an objective and implicit biomarker for diagnosing dopamine-related and neurodevelopmental disorders.

## Results

### Eye blinks spontaneously synchronize with musical beats

In Experiment 1, 30 nonmusicians listened to 10 Bach chorales at 85 beats per minute in both the original and reverse versions, while EEG and eye-tracking data were recorded simultaneously. In the reverse version, the sequence of beats was inverted from end to beginning while maintaining temporal regularity at the beat level and the original's acoustic properties (Fig 1B and 1C, see Supporting information and S1 Fig for details). This design ensured that the reverse version served as a control for typical tonal harmonic progressions, in which the novel harmonic progressions were less familiar to the participants. Each musical piece was repeated three times consecutively to collect sufficient data on a single piece and measure the effect of repetition. The experimental procedure is summarized in Fig 1A and 1B. The acoustic modulation spectra of all musical pieces are shown in Fig 1C, demonstrating prominent frequency components of musical beat structures (beat rate: 1.416 Hz) across the two conditions. To ensure that the reversal manipulation of harmonic progressions did not affect participants' liking of the music, we compared the liking ratings for the original and reverse versions. No significant difference in liking ratings was found between the two versions ($t_{(29)} = 0.715$, $p = 0.480$, Cohen's $d = 0.131$; Fig 1D), suggesting that preferences were unlikely to account for any differences in blink or neural signals observed later.

Next, we measured blink synchronization with musical beats and neural entrainment to beats (Fig 1E). We quantified continuous eye-tracking data by converting it into binary time series to identify the timing of eye blinks. Fast Fourier transform was then applied to the blink signals, and amplitude spectra of eye blinking were derived. Similarly, we extracted music-related EEG components (see Materials and methods for details) and derived amplitude spectra to examine whether neural components corresponded to musical beats, a routine analysis of neural entrainment to musical rhythms.

To our surprise, while neural entrainment to musical beats has been well documented [38–43], a similar finding was shown in the spectra of eye blinking: eye blinks spontaneously synchronized with musical beats, even without any instructions to do so. The blink synchronization with musical beats was significant, as demonstrated by the spectral peaks of eye blinking dynamics exceeding the thresholds ($p < 0.01$) created through a surrogate test on the group-averaged data (Fig 2A). We further derived the corrected amplitude within the significant frequency range for further analyses, which spanned from 1.416 to 1.433 Hz.

This finding, though surprising and perhaps entirely new, was robustly observed across both versions of musical pieces and all three repetitions. This is further echoed by a histogram of raw intervals between two consecutive blinks (S2 Fig), which shows a clear peak of intervals between one and two beats.

To assess the effects of different versions of musical pieces and repetition on blink synchronization, we conducted a two-way repeated-measures ANOVA, with version (original or reverse) and repetition (1, 2, or 3) as main factors, on the corrected amplitude within the significant frequency range (Fig 2B). We did not find any significant main effects nor interaction

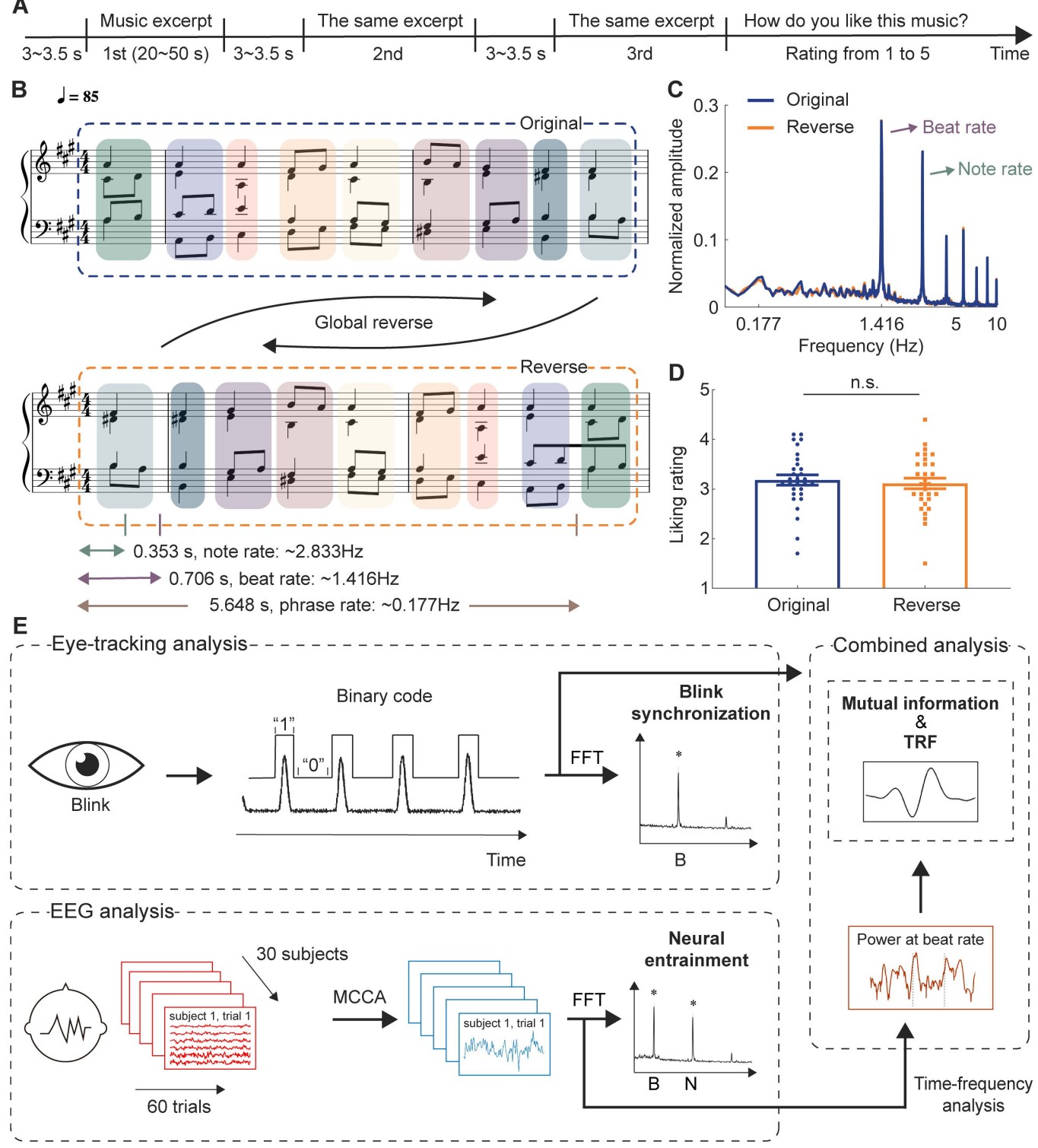

**Fig 1. Design and data analysis pipelines of Experiment 1. (A)** Each music excerpt was presented three times consecutively, during which EEG and eye-tracking data were recorded. After each trial, participants rated their liking of the excerpt. **(B)** Example of music excerpts. The tempo is 85 beats per minute: each note is ~0.353 s (2.833 Hz), each beat is ~0.706 s (1.416 Hz), and each phrase is ~5.648 s (0.177 Hz). The reverse version (bottom) was created by reversing the beat order of the original excerpts (top). **(C)** Modulation spectra of the stimuli showed peaks at the note and beat rates as

well as their harmonics for both versions. **(D)** Liking rate showed no significant difference between the two versions. **(E)** Overview of the analysis. Blinks were recorded via an eye tracker and converted into binary time series. The synchronization value around the beat rate (B) was measured by the fast Fourier transform (FFT) of the blink signals. EEG was recorded with 64 electrodes, and multiway canonical correlation analysis (MCCA) extracted the most common components across subjects. Denoised signals were used to measure neural entrainment to beats (B) and notes **(N)**. Neural signal power at the beat rate was obtained using a wavelet transform. Mutual information and temporal response function (TRF) measured the relationship between blink signals and neural power at the beat rate. Data are available on OSF (https://osf.io/xhfn4/). n.s., not significant.

(version: $F_{(1,26)}$ = 0.396, $p$ = 0.535, $\eta_p^2$ = 0.015; repetition: $F_{(2,52)}$ = 0.378, $p$ = 0.687, $\eta_p^2$ = 0.014; interaction: $F_{(2,52)}$ = 0.201, $p$ = 0.818, $\eta_p^2$ = 0.008). This lack of significance may stem from the fact that the beat structure was well preserved in different versions and was easy for participants to follow in each condition. As a result, even though blink synchronization to the beats was evident, no significant differences were found between repetitions or between the original and reverse versions.

We then tested whether blink synchronization performance correlates with musical ability measured by the rhythm subtest of the musical ear test (MET) [44]. The corrected amplitude within the significant frequency range were averaged across repetitions for each version and correlated with the MET rhythm score to examine the effect of musical rhythmic ability. No significant correlations were found for either version (original: $r_{(27)}$ = 0.071, $p$ = 0.723; reverse: $r_{(27)}$ = 0.169, $p$ = 0.399; Fig 2C). This lack of correlation is likely due to the fact that the beat structure in our stimuli was relatively easy to synchronize to, as noted earlier.

## Neural responses entrained to musical structures

In Experiment 1, we also replicated the established phenomenon of neural entrainment to musical beats. In line with previous studies [38–43], robust neural entrainment to beats was evident (Fig 2D). After preprocessing raw EEG data and removing independent components associated with eye blinks, eye movements, and heartbeat using an independent component analysis (ICA) algorithm, we applied multiway canonical correlation analysis (MCCA) to extract components linked to music listening (see Materials and methods). This approach enabled the extraction of auditory neural signals specific to music listening and facilitated the analysis of single-trial data for each musical piece and repetition. We then derived amplitude spectra of neural signals corresponding to music listening, and conducted the surrogate test to extract the corrected amplitude within the significant frequency ranges (at beat rate, 1.383–1.45 Hz; at note rate, 2.8–2.9 Hz) for both beat and note rates for further analyses.

To examine the effects of version and repetition, we performed a two-way repeated-measures ANOVA for beat and note rates separately (S3 Fig). At the note rate, the analysis showed no main effect of version ($F_{(1,26)}$ = 0.686, $p$ = 0.415, $\eta_p^2$ = 0.026) nor interaction ($F_{(2,52)}$ = 0.845, $p$ = 0.435, $\eta_p^2$ = 0.031), but a main effect of repetition ($F_{(2,52)}$ = 4.324, $p$ = 0.018, $\eta_p^2$ = 0.143), with larger amplitude during the second presentation than the first presentation ($p$ = 0.029, Bonferroni-corrected). At the beat rate, neural tracking of beats was unaffected by either factor or their interaction (version: $F_{(1,26)}$ = 0.440, $p$ = 0.513, $\eta_p^2$ = 0.017; repetition: $F_{(2,52)}$ = 0.908, $p$ = 0.409, $\eta_p^2$ = 0.034; interaction: $F_{(2,52)}$ = 1.416, $p$ = 0.252, $\eta_p^2$ = 0.052). Additionally, neural entrainment to notes and beats correlated with the MET rhythm score solely for the reverse version (note rate: $r_{(27)}$ = 0.436, $p$ = 0.023; beat rate: $r_{(27)}$ = 0.417, $p$ = 0.030). This suggests that although blink synchronization to the beat did not differ between the two versions due to their identical beat structures (Fig 1C), higher musical ability may be required to effectively track beat structures in the reverse version for neural entrainment, possibly due to the unfamiliar harmonic progressions (S1D Fig). Consequently, the reverse version demands enhanced rhythmic skills to accurately follow the music's beat patterns upon first exposure.

## Neural entrainment to musical beats corresponds to eye blinks

Subsequently, in Experiment 1, we examined the correspondence between blink synchronization and neural entrainment to beats. It is shown that the strength of neural entrainment correlates with behavioral measures of sensorimotor synchronization skills [45]. Extending this, the current study discovered that blink synchronization performance was significantly correlated with neural

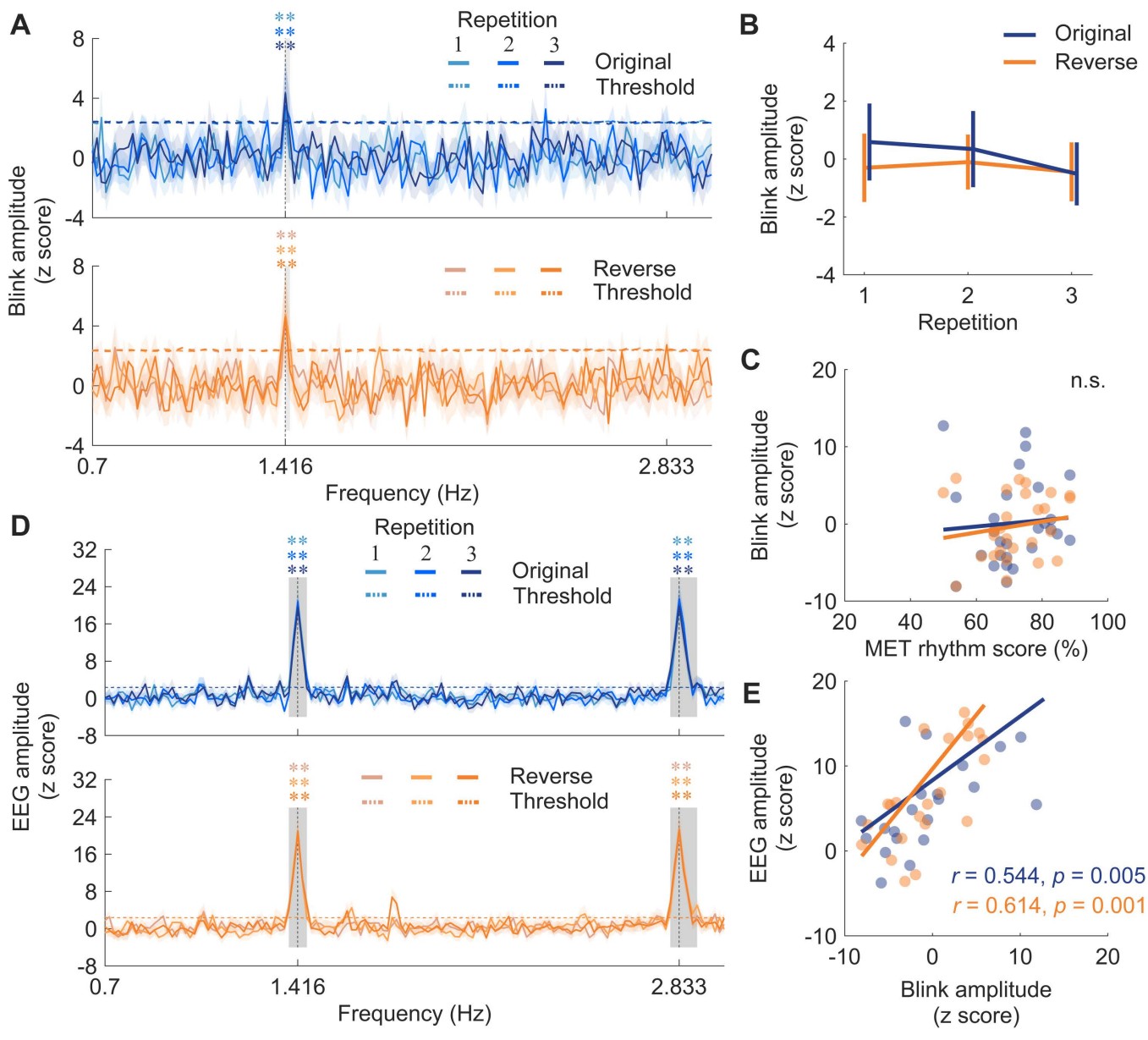

**Fig 2. Blink and neural synchronization to musical beats. (A)** Blink amplitude spectra for the two versions and repetitions, with horizontal dashed lines showing the threshold from the surrogate test ($p < 0.01$). Shaded areas represent ± one SEM across participants ($n = 30$). Gray boxes indicate the frequency ranges where amplitude was above the threshold. The blink amplitude showed a salient peak around the beat rate. **(B)** Blink amplitude within significant frequency ranges around the beat rate, with error bars denoting ± one SEM. Neither version nor repetition had a significant effect on blink amplitude. **(C)** Correlation between the musical ear test (MET) rhythm score and blink amplitude, showing no significant relation. Colored dots represent individuals. **(D)** Neural amplitude spectra for the two versions, with shaded areas representing ± one SEM ($n = 30$). Horizontal dashed lines represent surrogate test thresholds ($p < 0.01$). Gray boxes indicate the significant frequency ranges. Salient peaks appeared at the note and beat rates for both versions. **(E)** Partial correlation between blink amplitude and neural amplitude at the beat rate. Blink amplitude was positively correlated with neural amplitude for both versions. Colored dots represent individuals. Data are available on OSF (https://osf.io/xhfn4/). ** $p < 0.01$; n.s., not significant.

entrainment, after controlling for musical rhythmic ability as measured by the MET rhythm score (original: $r_{(23)}$ = 0.544, $p$ = 0.005; reverse: $r_{(23)}$ = 0.614, $p$ = 0.001; Fig 2E). Given the distinct scalp topographies of the first MCCA neural component and the eye-related components (including blinks and horizontal saccades) extracted via ICA (S4A Fig), which demonstrate clear spatial dissociation between neural and ocular sources, the observed blink-neural relationship cannot simply be attributed to blink artifacts.

To further investigate the underlying mechanism, we examined their relationship through mutual information (MI) and eye-blink-triggered temporal response function (TRF). MI between eye blink timing series and neural signals across entire musical pieces indicates whether the brain encodes information from the blinks. Given the blink–neural relationship was specific to the beat rate, we conducted a wavelet transform on EEG signals to derive the power of neural responses at the beat frequency, and then calculated the MI between the EEG power and blink signals. Indeed, compared to surrogate data, empirical MI values were significantly larger for all repetitions in the original version ($ps < 0.01$; Fig 3A). However, the reverse version yielded a significant result only in the third presentation ($p$ = 0.466, $p$ = 0.100, and $p$ = 0.004 for the first, second, and third presentations, respectively). A two-way repeated-measures ANOVA revealed a significant main effect of version ($F_{(1,27)}$ = 4.396, $p$ = 0.046, $\eta_p^2$ = 0.140; Fig 3B), showing higher MI in the original version. It also revealed a significant main effect of repetition ($F_{(1.629,43.979)}$ = 9.767, $p$ = 0.001, $\eta_p^2$ = 0.266), with higher MI on the second and third presentations than the first ($p$ = 0.002 and $p$ = 0.010, second and third presentations respectively, Bonferroni-corrected). The interaction was not significant ($F_{(2,54)}$ = 1.479, $p$ = 0.237, $\eta_p^2$ = 0.052).

Interestingly, MI results suggest that repetitions or listeners' familiarity with musical pieces modulated neural coupling to eye blinks, despite no such effect being observed on either neural entrainment (S3 Fig) or blink synchronization (Fig 2B), individually. This dependency was evident across all repetitions of the original version but only emerged during the third repetition of the reverse version. This discrepancy between the two versions, despite their identical beat structures, implies that effective prediction—facilitated by familiar harmonic progressions in the original version (S1D Fig)—may strengthen the coupling between blink and neural responses. Furthermore, the repetition effect underscores that blink synchronization transcends mere reflexive responses to sensory stimulation.

## Neural entrainment to musical beats is time-locked to blink onset

Next, in Experiment 1, we investigated how the brain encodes eye blinks over time. Employing TRF methodology, we extracted blink onset as the specific feature and examined the fluctuations in neural power at the beat rate before and after eye blinks, as illustrated in Fig 3C. Remarkably, neural responses were time-locked to blink onset, with an increase in neural responses, as reflected in TRF weights, prior to blink onset (Fig 3D). Specifically, for the original version, TRF weights were significantly higher than the surrogate-derived threshold before blink onset during the second and third presentations. Similarly, for the reverse version, TRF weights exceeded the threshold before blink onset during the third presentation. The repetition effect observed here echoes the MI results. A two-way repeated-measures ANOVA on the root mean square (RMS) of TRF weights at the response peak found a significant main effect of version ($F_{(1,27)}$ = 10.420, $p$ = 0.003, $\eta_p^2$ = 0.278; Fig 3E), suggesting higher RMS of weights for the original version. It also showed a significant main effect of repetition ($F_{(2,54)}$ = 4.799, $p$ = 0.012, $\eta_p^2$ = 0.151), with higher weights during the third presentation than the first presentation ($p$ = 0.004, Bonferroni-corrected), indicating a gradual formation of prediction for blink onset in the brain.

Thus far, we have established that blink synchronization—wherein eye blinks track musical beats—robustly corresponds with neural entrainment to beats. Moreover, blink synchronization transcends mere passive reflex to sensory stimuli, and the blink-neural relation is modulated by experimental factors such as repetition and reversal manipulations.

## Microstructure of the left posterior superior longitudinal fasciculus is associated with blink synchronization

Previous studies have revealed notable individual differences in auditory-motor synchronization abilities, which are linked to variations in white matter microstructural properties of sensorimotor pathways [46–48]. In Experiment 1, we examined the relationship between white matter microstructural properties and blink synchronization/neural entrainment performance

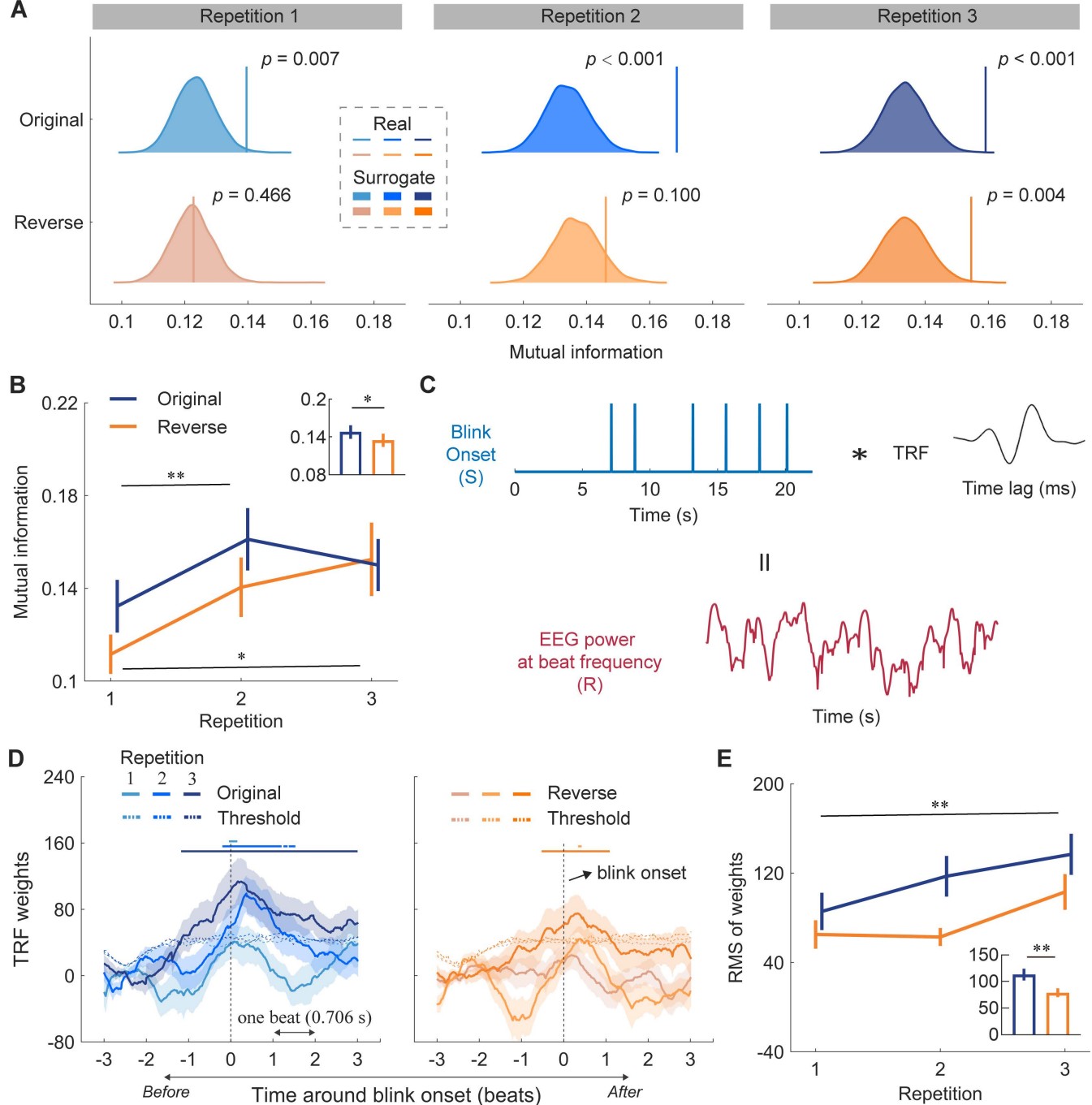

**Fig 3. Correspondence between blinks and neural signals. (A)** Mutual information (MI) between blink signals and beat-rate EEG power. Colored distributions represent surrogate MI values derived from circularly shifted EEG data. Colored vertical lines indicate the observed MI values. **(B)** Averaged MI for each version and repetition. The bar plot shows the mean MI values for both versions. Error bars denote ± one SEM. MI increased with repetition and was higher in the original version. **(C)** Temporal response function (TRF) analysis of EEG and eye blinks. Blink onsets were extracted from the blink signals and regressed against the EEG signals, resulting in the blink onset TRF. **(D)** TRF of beat-rate EEG power using blink onset as the regressor. Shaded areas represent ± one SEM across participants ($n$ = 30). Vertical dashed lines mark blink onset. Horizontal axis covers three beats before and after blink onset. Horizontal dashed lines show permutation test thresholds ($p < 0.01$). Horizontal solid lines at the top indicate significance. Notably, TRF weights increased significantly before blink onset. **(E)** Root mean square (RMS) of TRF weights at the peak. The bar plot shows the mean TRF weights for both versions. Error bars denote ± one SEM. RMS of weights was larger in the original version. Data are available on OSF (https://osf.io/xhfn4/). * $p < 0.05$, ** $p < 0.01$.

to further explore the structural variability associated with auditory-motor synchronization. Previous studies have found a positive correlation between fractional anisotropy (FA) of the left (but not right) arcuate fasciculus (AF) and auditory-motor synchronization [46–48]. In addition to the AF, the superior longitudinal fasciculus (SLF), which connects auditory and dorsal premotor regions through parietal regions, has also been implicated in auditory-motor synchronization and beat perception [18]. Based on these findings, we hypothesized that individuals with better blink synchronization or neural entrainment would exhibit stronger structural connectivity in the left AF and SLF, or greater left-lateralization of these tracts.

Firstly, we acquired diffusion-weighted imaging (DWI) data to quantify potential differences in white matter tracts connecting frontal, parietal, and auditory regions which are necessary for auditory-motor synchronization and beat perception. We extracted three microstructural indices—FA from the diffusion tensor imaging (DTI) model, neurite density index (NDI) and orientation dispersion index (ODI) from the neurite orientation dispersion and density imaging (NODDI) model—as well as their lateralization indices (LI) for bilateral SLF and AF segments to explore the relationship between blink synchronization/neural entrainment and white matter properties (see Fiber tractography in Materials and methods). Partial correlation analyses controlling for musical rhythmic ability revealed a significant negative correlation between blink amplitude and the NDI of the left posterior SLF (pSLF) ($r_{(22)}$ = −0.540, $p$ = 0.048, Bonferroni-corrected, Fig 4A). Neural entrainment was also significantly and negatively correlated with the NDI of the left AF ($r_{(22)}$ = −0.560, $p$ = 0.032, Bonferroni-corrected, Fig 4B). These findings suggest that better blink and neural synchronization may be associated with lower neurite density in the left pSLF and AF. No significant correlations were found for other microstructural indices (see S2 Table for full statistics).

## Eye blinks are primarily entrained by temporal patterns in music

Despite our confirmation of this novel behavior, several questions remain, particularly regarding which musical elements drive this behavior. Specifically, does blink synchronization depend on pitch information or melodic patterns, or is it solely

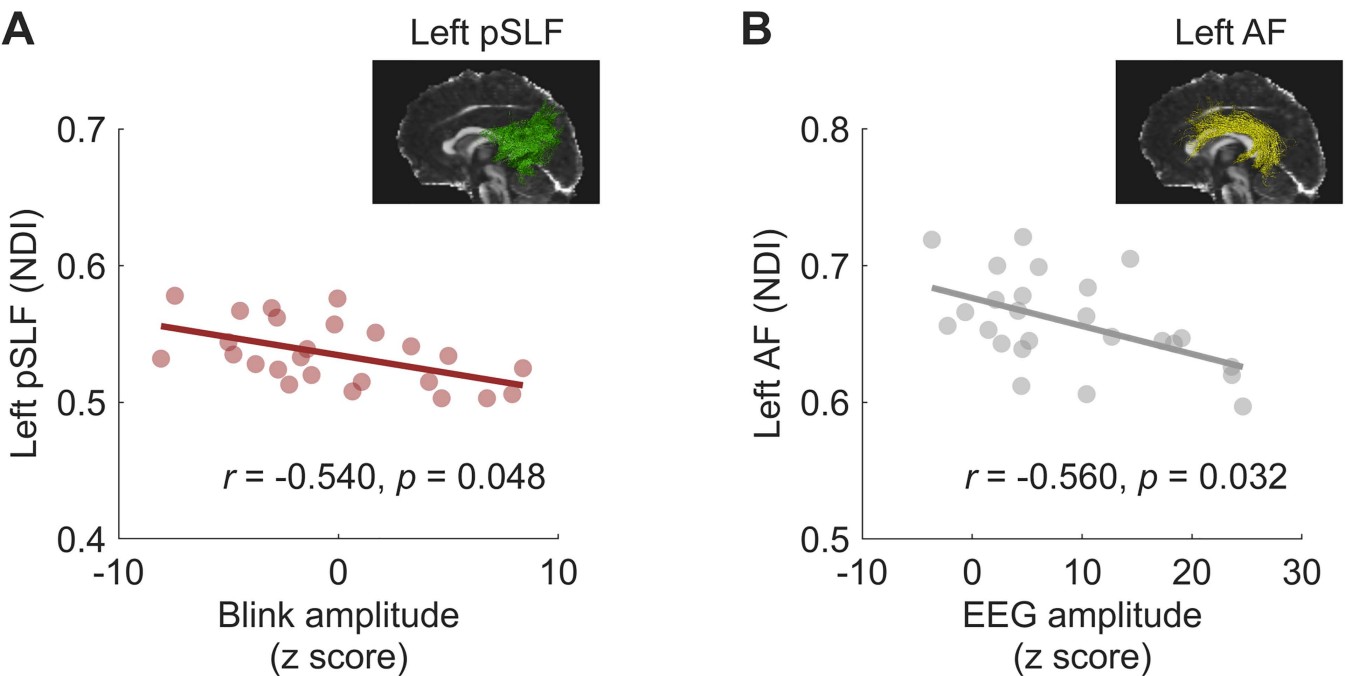

**Fig 4. Partial correlations between blink/neural synchronization and white matter microstructure, controlling for musical rhythmic ability. (A)** Blink amplitude averaged across the original and reverse versions was negatively correlated with the neurite density index (NDI) of the left posterior superior longitudinal fasciculus (pSLF). **(B)** EEG amplitude at the beat rate averaged across the two versions was negatively correlated with the NDI of the left arcuate fasciculus (AF). Colored dots represent individuals. Data are available on OSF (https://osf.io/xhfn4/).

driven by temporal patterns in tone sequences deprived of melodic cues? Additionally, given that rhythmic rate has been shown to influence people's abilities to synchronize with auditory-motor activities, such as tapping, whispering, or clapping in synchrony with auditory stimuli [49,50], does musical tempo also modulate blink synchronization?

To address these questions, in Experiment 2, we compared blink synchronization to both music and static tone sequences across three tempi (66, 85, and 120 beats per minute, corresponding to inter-beat intervals of 0.909 s, 0.706 s, and 0.5 s, respectively) (Fig 5A). We generated 10 new tone sequences maintaining the temporal structures of the original musical pieces but lacking pitch information or melodic cues. Thirty nonmusicians listened attentively to each auditory stimulus and pressed the spacebar as soon as possible upon detecting a timbre deviant, while their eye-tracking data were recorded. Trials containing timbre deviants were excluded from analyses.

A key difference from Experiment 1 was the inclusion of a 3-minute rest period before the main experiment to determine whether eye blinks during idle state exhibit any temporal structures. A surrogate test on the baseline data yielded no significant peak (Fig 5B), confirming that the spectral peak observed in Experiment 1 was indeed music-related or at least stimulus-related.

In Fig 5C, we replicated the findings in Experiment 1, observing robust blink synchronization in Experiment 2. Importantly, musical tempo modulated blink synchronization, with spectral peaks aligning with the beat rates across stimulus types and tempi, except at the highest tempo (120 beats per minute) for original pieces. We extracted the corrected amplitude within the significant frequency ranges derived from the surrogate test for the three tempi: 66 beats per minute, 1.100 Hz; 85 beats per minute, 1.416 Hz; and 120 beats per minute, 2 Hz. Then we tested the effects of stimuli type (original pieces or tone sequences) and tempo (66, 85, or 120 beats per minute) using a two-way repeated-measures ANOVA. Although blink synchronization appeared stronger for original pieces than tone sequences and decreased with increasing tempo, we found no significant main effects (stimuli type: $F_{(1,28)} = 0.664$, $p = 0.422$, $\eta_p^2 = 0.023$; tempo: $F_{(2,56)} = 0.416$, $p = 0.662$, $\eta_p^2 = 0.015$) or interaction effect ($F_{(2,56)} = 0.193$, $p = 0.825$, $\eta_p^2 = 0.007$). Moreover, there was no significant correlation between blink synchronization and the MET rhythm score (original pieces: $r_{(29)} = -0.287$, $p = 0.131$; tone sequences: $r_{(29)} = -0.337$, $p = 0.073$; Fig 5D).

Taken together, the results in Experiments 1 and 2 suggest that blink synchronization is primarily entrained by the temporal patterns in music, rather than pitch changes or melodic cues. Eye blinks align with the timing of auditory events, supporting the idea that temporal structure is the key driver of synchronization.

## Blink synchronization facilitates pitch deviant detection

In Experiment 3, we investigated whether blink synchronization correlates with behavioral performance in a detection task. While prior experiments have established a connection between blink synchronization and neural entrainment to beats, we further sought to determine if better blink synchronization corresponds to improved tracking of musical elements? To do so, we conducted a pitch deviant detection task where nonmusicians ($n = 31$) listened to music segments selected from the original pieces. Each segment contained two phrases (16 beats in total), with a pitch deviant introduced in the second phrase. Participants were instructed to detect deviant tones as soon as possible, while their eye-tracking data were recorded (Fig 6A).

We replicated the signature of blink synchronization to beats observed in earlier experiments (Fig 6B). We further found a positive correlation between blink synchronization strength and deviant detection accuracy ($r_{(30)} = 0.543$, $p = 0.002$; Fig 6C left). However, no significant association was observed between blink synchronization and reaction time ($r_{(31)} = -0.165$, $p = 0.374$; Fig 6C right). These results suggest that proficient blink synchronization may facilitate the detection of deviant pitch by aligning temporal attention with beat onset, implying a potential role of blink synchronization in a dynamic attending process.

## Blink synchronization disappeared when music is task-irrelevant

In Experiment 3, we suggested that blink synchronization reflects a dynamic attending process. Given that one of the functions of eye blinks is to regulate the flow of visual information, we next investigated blink synchronization in a

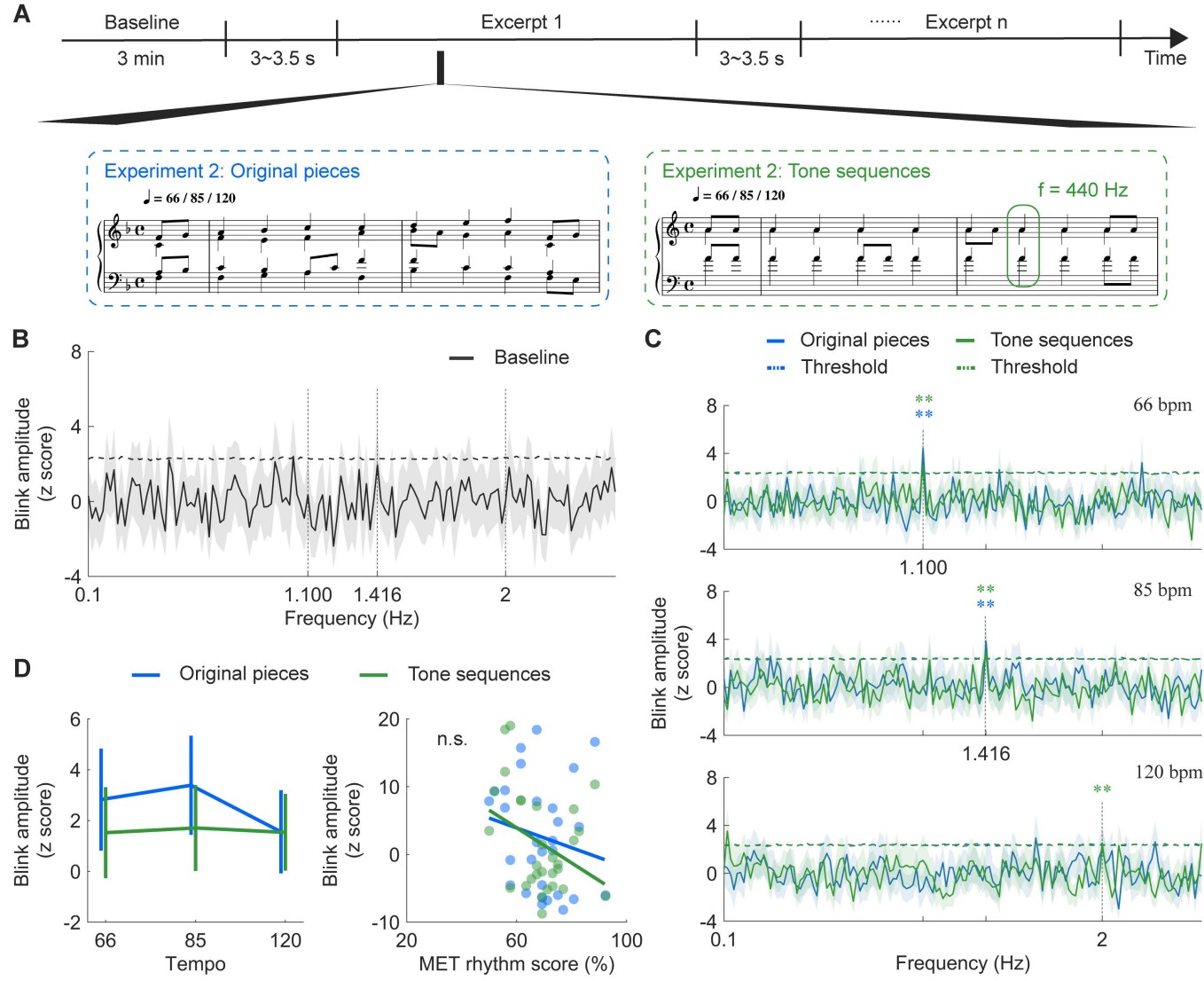

**Fig 5. Blink synchronization in response to rhythmic sequences. (A)** Experimental paradigm for Experiment 2: participants listened to both original musical pieces and tone sequences while their eye-tracking data were recorded. **(B)** Blink amplitude spectra for the baseline (no music), with horizontal dashed lines representing the surrogate test threshold ($p < 0.01$). Shaded areas represent ± one SEM across participants ($n = 30$). **(C)** Blink amplitude spectra for two stimuli types at three tempi, with horizontal dashed lines showing the surrogate test threshold ($p < 0.01$). Vertical dashed lines indicate the frequency points corresponding to the beat rates. Shaded areas represent ± one SEM. Blink amplitude showed salient peaks at the beat rate across stimuli types and tempi, except at 120 beats per minute (bpm) for original pieces. **(D)** Left: Blink amplitude at the beat rate. Error bars denote ± one SEM. Stimuli type and tempo had no significant effect on blink responses. Right: No significant correlation was found between musical ear test (MET) rhythm score and blink amplitude. Colored dots indicate individuals. Data are available on OSF (https://osf.io/xhfn4/). ** $p < 0.01$; f, frequency; n.s., not significant.

cross-modal context in Experiment 4. The experimental procedure mirrored that of Experiment 3, except that 32 nonmusician participants were required to detect a simple visual target—a red dot—presented on the screen while listening to music (Fig 6A). The deliberate simplicity of the visual task was designed to ensure participants' attention primarily towards the music. We manipulated the temporal relationship between the visual target and musical beat onsets, resulting in two conditions: on-beat and off-beat. If blink synchronization reflects the modulation of visual sampling efficiency by musical

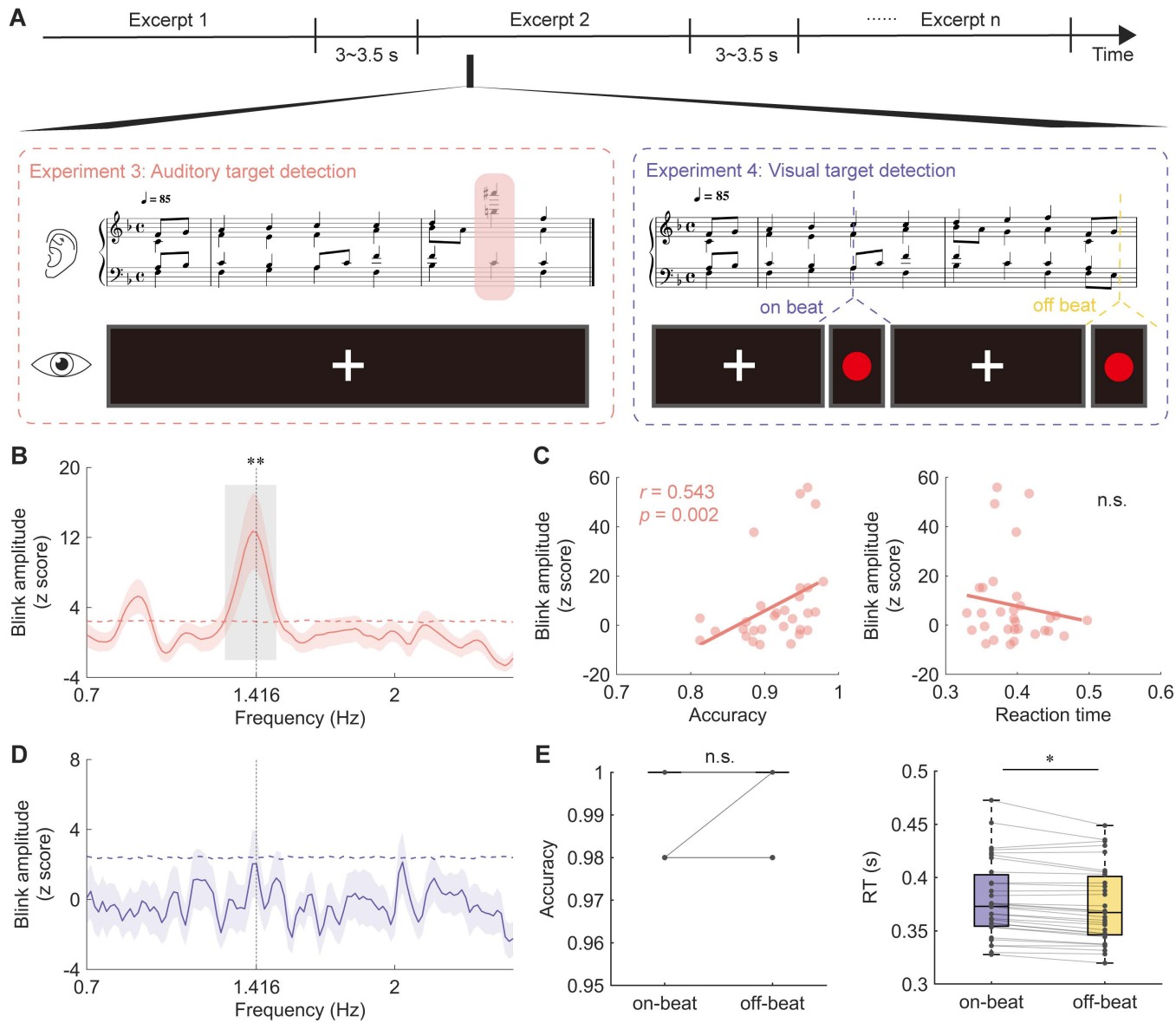

**Fig 6. Functional significance of blink synchronization in behavioral performance. (A)** Experimental paradigm for Experiments 3 and 4: participants performed a pitch deviant detection task (Experiment 3) or a visual target detection task (Experiment 4) while their eye-tracking data were recorded. **(B)** Blink amplitude spectra in Experiment 3, with horizontal dashed line representing the surrogate test threshold ($p < 0.01$). Shaded area represents ± one SEM across participants ($n = 31$). The gray box indicates the frequency range above the threshold. The blink amplitude showed a salient peak around the beat rate. **(C)** Correlation between blink amplitude and behavioral indexes in Experiment 3. Blink amplitude was positively correlated with detection accuracy, but not with reaction time. Colored dots indicate individuals. **(D)** Blink amplitude spectra in Experiment 4 ($n = 32$), with no significant peak observed. **(E)** Group-averaged accuracy and reaction time in the on-beat (purple) and off-beat (yellow) conditions. Colored connecting lines represent individual participants. Participants responded significantly faster to off-beat visual target than to on-beat target. Data are available on OSF (https://osf.io/xhfn4/). * $p < 0.05$, ** $p < 0.01$; n.s., not significant.

rhythms, we would expect to observe a correlation between blink synchronization and behavioral performance as found in Experiment 3, and better performance, like higher accuracy or shorter reaction time, in the on-beat condition than the off-beat condition.

Despite these expectations, eye blinks did not even synchronize with musical beats in this task ([Fig 6D]). This observation is particularly noteworthy, as participants were not heavily engaged in the visual task, as evidenced by nearly 100% accuracy in both conditions ($t_{(27)}$ = −0.812, $p$ = 0.424, Cohen's $d$ = −0.153, [Fig 6E] left). A plausible explanation is that blink synchronization is modulated by task modality and the allocation of auditory attention, in which the visual task itself may interfere with the eye blinks and also divert attention from the auditory stream. We further compared reaction time for detecting the visual target between on-beat and off-beat conditions using a two-sided paired $t$ test. Unexpectedly, responses were significantly slower in the on-beat condition ($t_{(30)}$ = 2.417, $p$ = 0.022, Cohen's $d$ = 0.434, [Fig 6E] right).

## Discussion

We've demonstrated through a series of experiments that oculomotor activity, specifically eye blinking, spontaneously aligns with regular musical beats during active music listening—a phenomenon we term blink synchronization, and explored its neural substrates and functional significance. As depicted in [Fig 2], we discovered that eye blinks spontaneously synchronize with musical beats at 85 beats per minute. Moreover, we observed a correspondence between blink synchronization and neural entrainment to beats ([Fig 3]). On a structural level, we found that the microstructural connectivity of the left pSLF played an important role in the strength of blink synchronization ([Fig 4]). We then replicated this blink synchronization phenomenon across a broader range of tempi (66 and 85 beats per minute) of the original musical pieces and all three tempi (66/85/120 beats per minute) in tone sequences without melodic cues ([Fig 5]), affirming its robustness. Furthermore, our investigation into the functional relevance of blink synchronization revealed intriguing findings. Experiment 3 revealed a correlation between stronger blink synchronization and improved performance on a pitch deviant detection task in a musical context. Experiment 4 using a visual detection task highlighted the impact of task relevance on blink synchronization to music ([Fig 6]).

The spontaneous blink rate has long been recognized as a vital indicator of cognitive and neural functions [51]. Recent research has begun to explore the link between eye blinks and music listening, aiming to uncover the cognitive complexities of human music processing related to emotion [32], attention [33], and subjective states [34]. However, a direct connection between blinks and beat perception remains elusive. While motor behaviors such as finger tapping or dancing are commonly studied to understand musical rhythm and beat perception, an intriguing question remained: can music implicitly entrain blink activity? Our discovery of blink synchronization established a direct link between eye blinks and musical beat perception in a naturalistic listening setting, introducing a novel type of spontaneous music-synchronizing behavior. This is especially interesting because when people tap their feet or nod their heads in sync with the music, they are typically aware of it. In contrast, although we did not assess participants' self-awareness of their blinking behavior, none reported consciously blinking to the beat after the experiment. The fact that autonomic oculomotor activity can track musical beat may reflect the evolution-rooted primitive instinct in music rhythm processing [18].

In addition to eye blinks, prior studies have demonstrated that pupil size is modulated by microtiming asynchronies between musical beats [52], linking pupil dynamics to temporal attention and specific note configurations [53,54], and more recently, directly to musical beat structure [30]. In other auditory contexts, both eye blinks and pupil size have been shown to track hierarchical linguistic or nonlinguistic structures, reflecting a modulation of temporal attention [29]. Future research should investigate whether blink synchronization extends to higher-level musical structures, such as meter or phrase, and elucidate the relationship between blink activity and pupil dynamics. This helps to clarify the underlying cognitive processes and mechanisms involved, and differentiate oculomotor synchronization in music from that in speech processing.

Our results also provide compelling evidence for brain–eye coordination in beat perception, with blink activity and neural entrainment activity coupled during music listening ([Fig 3A] and [3B]). Specifically, eye blinks were represented in cortical responses, with a time-locked neural response gradually preceding blink onset across the first to third repetitions ([Fig 3D] and [3E]), suggesting a refinement of neural prediction for upcoming blinks through short-term exposure. Aligning

with frameworks such as predictive coding [16,17], active sensing [14,15], and the ASAP hypothesis [18], our findings suggest that cortical oscillations, potentially originating from dorsal auditory-motor pathways, not only synchronize with incoming musical beats but may also contribute to the predictive timing of blink activity. By comparing the expected and actual timing of blinks across repetitions, the brain may continuously update its temporal predictions, leading to improved synchronization between blinks and neural signals. This enhanced synchronization may, in turn, facilitate more efficient auditory beat processing.

Moreover, we found that the microstructural integrity, indexed by NDI, of the left pSLF served as the structural basis for blink synchronization. The pSLF connects the auditory cortex and inferior parietal lobule encompassing the intraparietal sulcus, where the parietal eye field (PEF) is located [55]. Lower NDI, which reflects reduced neurite density, was associated with better blink synchronization performance (Fig 4A). Notably, NDI derived from the NODDI model provides a more precise and sensitive estimate of neurite morphology than FA [56]. A series of studies have similarly reported that lower neurite density in white and gray matter at the whole-brain level is linked to higher intelligence, better reading and phonological skills, and improved cognitive performance [57–59]. Consistent with these findings, our results suggest that lower NDI in the left pSLF and AF supports higher auditory-motor synchrony ability. This may reflect greater neural efficiency from synaptic pruning within these left-hemisphere tracts, rather than hemispheric asymmetry per se, thereby underscoring their unique role in the effective processing of musical beats. However, given the relatively small sample size, replication and stability assessment in larger cohorts are necessary. Furthermore, we did not find significant associations between blink synchronization and the structural properties of the dorsal SLF (dSLF), which connects the PEF to the dorsolateral frontal area encompassing the frontal eye field (FEF). This null finding may stem from limited statistical power due to the modest sample size and/or high homogeneity of musical ability within our participant group. Future studies incorporating larger and more diverse populations with a broader range of musical expertise will be essential for fully addressing this question.

Notably, we did not find significant blink synchronization at a tempo of 120 beats per minute for the musical stimuli (Fig 5C). This implies a rate limit for blink synchronization, a phenomenon also observed in finger tapping and covert synchronization studies [50,60]. Given differentially preferred tempo of rhythmic behaviors among motor effectors [49], further research is required to determine the optimal tempo range for robust blink synchronization to musical beats. Interestingly, blink synchronization was restored when original musical pieces were replaced with tone sequences at 120 beats per minute. This result is consistent with a recent study showing that repeated acoustic units facilitate auditory-motor synchronization in a subgroup of the population [61], inspiring us to take acoustic features into consideration in future experiments.

In Experiment 3, we found that blink synchronization performance was positively correlated with the detection accuracy of pitch deviant that appeared at the anticipated beat (Fig 6C), consistent with previous findings [19,20,62]. This result supports the DAT [35–37] and the active sensing model [14,15], suggesting that, similar to other motor behaviors, synchronized blinks help entrain attention to musical beats, enhancing the precision of temporal prediction, thereby facilitating the detection of pitch deviants. However, in Experiment 4, blink synchronization disappeared when the music was task-irrelevant (Fig 6D), suggesting that auditory rhythms only entrain blinks when they are task-relevant and well-attended. A prior study has demonstrated that gaze aligns with speech dynamics and the strength is modulated by selective attention [63]. In our task, participants were instructed to attend to visual stimuli, which may have inhibited the processing of auditory information and contributed to the absence of blink synchronization. Moreover, the visual task demands may have induced strategic blink inhibition to prioritize visual perception, thereby muting the effect of music on blink synchronization. A potential ceiling effect in task performance may also have obscured active sensing mechanisms. Future studies are needed to investigate cross-modal temporal modulation of blink synchronization using more challenging tasks and modality-matched control tasks, such as tactile tasks.

The delayed response to on-beat visual target (Fig 6E) was notable and seemingly contradictory to the DAT [35–37], which suggests that periodic auditory rhythms entrain attention to specific time points to optimize perception, yielding

faster reaction times for stimuli presented in synchrony with the rhythm. Our results instead suggest a cross-modal interaction in dynamic attending. When visual targets appeared on the beat, participants may have focused more on the auditory beat event, delaying visual responses. In contrast, the off-beat condition likely allowed for more flexible attention allocation, leading to faster responses. Moreover, while most studies show that unattended musical rhythms facilitate visual perception and memory [64–66], others find that unattended musical rhythms either impair or have no effect on perception [67–69]. This disparity is likely attributable to several factors. First, most studies supporting rhythmic effects employed simple geometric shapes or objects as stimuli, with tasks requiring the detection or discrimination of low-level perceptual features, unlike the pseudowords used in studies yielding null findings in which required lexical-semantic integration, introducing competitive resource allocation that may override rhythmic effects. Second, the temporal relationship between beats and off-beat targets significantly influences outcomes. While off-beat targets in many prior studies appeared at fixed intervals with respect to the beat, in both the current study and previous studies reporting null effects, off-beat targets were randomly distributed relative to the beat and may impair the precision of temporal prediction. Finally, methodological differences in experimental design may have contributed to the inconsistent findings. Studies reporting rhythmic effects typically used blocked designs, whereas our study employed an intermixed design in which on-beat and off-beat targets occurred within the same block. This design likely weakened the temporal expectations induced by the musical rhythm, thereby diminishing the rhythmic effect on target detection. Our result may also reflect a post-blink boost effect, where blinks help reformat visual information and enhance perceptual sensitivity immediately after blinks, as demonstrated in previous research [70,71]. As illustrated in S5 Fig, eye blinks were more likely to occur between two beats during the visual detection task in Experiment 4, potentially facilitating the perceptual processing of off-beat targets.

Musical beat perception likely engages a subcortical-cortical interaction between the basal ganglia and premotor areas [72]. While some studies have linked the premotor areas to peripheral oculomotor activities in nonhuman animals [73,74], other studies on eyeblink conditioning have identified a close relationship between the basal ganglia and eye blinks [75,76]. It is plausible that the basal ganglia, stimulated by musical beats, signals the FEF and oculomotor muscles, prompting eye blinks via a subcortical-cortical-peripheral pathway. This echoes the finding shown in Fig 3D, where neural entrainment to beats predicted eye blinking activities. Nevertheless, the neural circuit underlying this phenomenon needs future investigations.

The discovery of blink synchronization marks a significant stride in our comprehension of auditory-motor synchronization, yet this study has certain limitations that present avenues for future exploration. First, the relatively modest sample size, comprising primarily of young nonmusicians with normal hearing, should be expanded to include larger samples and more diverse populations. Including musicians can reveal how musical training influences blink synchronization, while diverse age groups could help assess developmental trajectories. Given that impaired rhythm processing is a risk factor for neurodevelopmental disorders [4], blink synchronization during music listening may have great potential to serve as an implicit and feasible diagnostic and interventional biomarkers for children, compared to explicit paradigms like finger tapping. Second, the exclusive use of Bach's chorales, noted for high temporal regularity and representation of Western classical music, suggests a limitation in musical diversity. Investigating blink synchronization in response to music featuring irregular rhythms, syncopation, and a variety of musical genres and cultural backgrounds is crucial for understanding the universality of this phenomenon and its dependency on rhythmic structures. Furthermore, given that pupil dynamics [30,52–54] and saccades [31] are found to be implicated in processing auditory rhythms and musical structures, the coordination between blinks, pupil size, and saccades in beat perception warrants further investigation to advance our understanding of eye movements in music listening. Finally, future neuroimaging research should examine the neural underpinnings of blink synchronization, particularly focusing on the basal ganglia, FEF, and premotor areas.

In conclusion, our study sheds light on the relationship between music listening and oculomotor activity, specifically the synchronization of eye blinks with musical rhythms. Our findings establish blink synchronization as a novel, distinctive, spontaneous auditory-motor synchronization behavior during music listening, and reveal its neurophysiological and

structural correlates. By linking active sensing and ASAP frameworks to blink synchronization, we gain deeper insights into the coordination of auditory, motor, and visual systems, thereby enriching our knowledge of cross-modal interaction and embodied musical perception.

## Materials and methods

### Ethics statement

All participants provided written consent before taking part in the study, which was approved by the Ethics Committee of the Institute of Psychology, Chinese Academy of Science (Approval No. H23070). The study has been conducted according to the principles expressed in the Declaration of Helsinki.

### Participants

A total of 123 young adults (70 women; mean age, 22.67 ± 2.98 years, ranging from 18 to 34 years old) with normal hearing (thresholds ≤20 dB SPL from 125 to 8,000 Hz) took part in the study. Thirty individuals participated in Experiment 1, 30 in Experiment 2, 31 in Experiment 3, and 32 in Experiment 4. The "musical training" subscale of the Goldsmiths Musical Sophistication Index (Gold-MSI) questionnaire [77] was used to evaluate the amount of musical training of participants, on a scale from 7: no training to 49: more training than 99% of the random sample. All participants scored an average of 17.45 (SD = ± 7.60, range: 7–39) on the musical training subscale and none of them was a musician. All participants reported normal or corrected-to-normal vision and an absence of neurological or psychiatric diseases.

### Stimuli

The stimuli comprised 10 musical pieces selected from the 371 four-part chorales by Johann Sebastian Bach (Breitkopf Edition, Nr. 8610). The original musical scores were checked manually to ensure inclusion of only quarter notes, eighth notes and 16th notes, while removing ties across notes to facilitate repetition. Importantly, we eliminated fermatas from the original musical pieces to prompt participants to parse the musical pieces by structural cues rather than rhythmical or acoustic cues for phrasal structures (such as pauses between phrases and lengthened notes). The duration of the musical pieces ranged from 22.59 to 59.30 s (39.82 ± 11.49 s, mean ± SD).

In Experiment 1, we created a reverse version by completely reversing the order of beats in each original piece (for an example piece, see Fig 1B). Therefore, the basic musical contents were equal in both versions, but the harmonic progressions were manipulated in the reverse version which would affect listeners' expectation or familiarity of the musical pieces (S1 Fig).

In Experiment 2, the original pieces were selected from eight of the 10 musical pieces used in Experiment 1, and the tone sequences consisted of frequent standard tones (440 Hz) with timing structures identical to the original pieces. To ensure participants' attention remained focused, we randomly inserted one to five chords deviating in timbre (guitar) into the remaining two musical pieces depending on the duration of piece (2.33 ± 1.44, mean ± SD). For the same piece, both the number of timbre deviants and their insertion positions varied across different stimuli type and tempo. These two pieces were interspersed within the block but excluded from further analyses.

In Experiment 3, the original versions of 10 musical pieces used in Experiment 1 were divided into 24 discrete segments, each comprising two complete phrases. Each phrase consisted of eight beats. A pitch deviant occurred randomly within the second phrase of each segment with equal probability. To adjust difficulty levels, we created pitch deviants by shifting the segment in the upper two voice parts (soprano and alto) upwards by one to two octaves. Each segment was manipulated into two versions with the pitch deviant inserted at different positions.

Experiment 4 utilized visual stimuli, with the target being a red circle with a diameter of 50 pixels (RGB: 255, 0, 0), displayed at the center on a black background (RGB: 0, 0, 0). The auditory stimuli employed in Experiment 4 were identical to those used as the original version in Experiment 1.

All stimuli were generated in MuseScore 3 and exported as wav files with a piano timbre, using the mechanical playback setting to remove expressive timing and dynamic cues. Musical excerpts were presented at a sampling rate of 44,100 Hz, and their intensity level were normalized to 70 dB SPL. Low-level acoustic analyses using Essentia 2.1 [78] confirmed the absence of expressive variation between conditions (S1E–S1H Fig). In Experiments 1, 3, and 4, musical stimuli were played at a tempo of 85 beats per minute, while Experiment 2 employed three tempi (66, 85, and 120 beats per minute). Therefore, the note rate for each musical piece at a tempo of 85 beats per minute was around 2.833 Hz, the beat rate was around 1.416 Hz, and the phrase rate was around 0.177 Hz.

## Experimental design and procedure

**Experiment 1.** The 20 musical pieces (10 pieces × 2 versions) were presented in 4 blocks, each containing 5 pieces. In each block, participants listened to each of the 5 pieces three times in a row, and the piece order was randomized for each participant. Throughout the experiment, participants were required to fixate on a white fixation cross at the center of a black screen. After the third presentation of each musical piece, participants pressed keys to rate how much they liked it using a 5-point Likert scale, with 1 indicating low preference and 5 indicating high preference. After providing their rating, the next musical piece commenced after a delay of 3–3.5 s (see Fig 1A). Each block lasted around 10 minutes and participants were allowed to rest between blocks.

**Experiment 2.** Different types of stimuli (original pieces or tone sequences) were presented in separate blocks, with the types alternating every 5 trials. Stimuli were presented to participants according to a balanced Latin square design for every type, and the order of stimulus types was counterbalanced across participants. Before the main experiment, participants underwent a 3-minute rest period while looking at a blank screen to establish a stable baseline for blink activity. During each block, participants were instructed to focus on the fixation cross displayed on the screen, attentively listen to the musical pieces, and press the spacebar as soon as possible upon detecting a timbre deviant. Participants received training until they achieved an accuracy level of 80%.

**Experiment 3.** Participants were presented with two versions of the 24 auditory segments, each presented twice, leading to 96 segments in total (24 segments × 2 versions × 2 times). Stimuli were presented in a pseudorandomized order to ensure that no segment or version was repeated consecutively. Participants were asked to carefully listen to the sequences and press the spacebar as soon as they detected a pitch deviant, while maintaining fixation on the central cross on the screen. Before the experiment, participants underwent a training session, concluding once they achieved an accuracy score above 80% on the detection task.

**Experiment 4.** Participants engaged in a visual target detection task while listening to 10 musical pieces—the same original versions used in Experiment 1. Each musical piece was presented twice in a pseudorandomized order. Concurrently, the fixation cross at the center of the screen occasionally transformed into a red circle (visual target). The visual target appeared either synchronously with the beat onset (on-beat condition) or randomly at one of nine temporal positions between two beats (off-beat condition). It was ensured that (1) each musical piece contained at least one and up to four visual targets per condition, depending on the duration of piece, with an equal number of on-beat and off-beat targets within each piece (2.50 ± 1.05, mean ± SD), and (2) no piece was presented twice consecutively with visual targets occurring at different positions for the same piece. The maximum duration of a red circle matched the musical beat interval (0.706 s), disappearing immediately upon spacebar response. Participants were instructed to observe the fixation cross while the auditory stimuli played, and press the spacebar as soon as possible once sighting the red circle on the screen. A training session was given before the experiment until participants achieved a target detection accuracy exceeding 80%.

At the end of each experiment, participants completed the Gold-MSI questionnaire and MET to measure their musical ability. The English version of the Gold-MSI questionnaire was translated into Chinese by the experimenters. Participants in Experiment 4 did not complete the MET due to the duration of the task.

## Blink acquisition and preprocessing (Experiment 1–4)

Eye-tracking data were recorded in all experiments using an Eyelink Portable Duo system (SR Research, Ontario, Canada), with a sampling rate of 500 Hz. Participants maintained a consistent eye-to-monitor distance of 95 cm, with the eye tracker positioned approximately 60 cm from their eyes. Before each experiment, a nine-point standard calibration and validation test was performed. Then a fixation cross appeared on the monitor with a resolution of 1,920 × 1,080 pixels. A re-calibration procedure was applied after each break to ensure accuracy and consistency.

For eye-tracking data, blink detection was the primary event of interest. Blinks were identified by the Eyelink tracker's on-line parser, which flagged instances where the pupil in the camera image was absent or severely distorted by eyelid occlusion. The time series were then divided into epochs corresponding to the duration of each musical or pure-tone sequence, aligning with the experimental trials. In Experiment 2, baseline data was derived from three continuous trials, each lasting 60 s. Blink durations shorter than 50 ms or longer than 500 ms were excluded from the analysis [79]. Subsequently, mean blink duration and blink rate were calculated for each participant. Blink rate was quantified as the number of blinks per minute for each trial and participant (units: blinks/min). Blinks with durations exceeding three standard deviations from the participant-specific mean across trials, and trials with blink rates exceeding three times the participant-specific standard deviation, were removed (see S3 Table for the mean and SD of the blink rate for each condition). Within each trial, data were represented as binary strings, with values set to 1 during blink occurrences and 0 otherwise. The inclusion of blink duration yields more nuanced temporal information compared to blink rate quantification, enabling a more comprehensive assessment of the dominant periodicity of temporal structure in the inherently discrete blink time series. Following preprocessing, eye-tracking data were down-sampled to 100 Hz for further analysis.

## EEG acquisition and preprocessing (Experiment 1)

EEG data were recorded using a 64-channel Quik-Cap based on the international 10–20 system (Neuroscan, VA, USA), with a sampling rate of 1,000 Hz. The EEG was referenced to the average of all electrodes, with skin/electrode impedance maintained below 5 kΩ. Additionally, four electrodes were used to record electrooculography (EOG): two electrodes were placed at the outer canthus of each eye, while another two electrodes were placed above and below the left eye to record the horizontal and vertical EOG signals, respectively.

For continuous EEG data, single trials were extracted spanning 3 s before stimulus onset to 3 s after stimulus offset. All data were band-pass filtered between 0.7 and 35 Hz using a phase-preserving two-pass fourth-order Butterworth filter, supplemented with a 48- to 52-Hz band-stop filter to eliminate line noise. Then the data were down-sampled to 100 Hz to match the sampling rate of the eye-tracking data. An ICA [80] with 30 principal components, implemented in Fieldtrip, was performed to remove EOG and electrocardiography (ECG) artifacts. Finally, we concatenated all trials into one matrix to derive the most common component of the EEG signals reflecting neural responses to musical pieces, using MCCA [81]. The first MCCA component was projected back to each participant and each trial, and used for subsequent analysis.

Preprocessing of eye-tracking and EEG data were performed in MATLAB (The MathWorks, Natick, MA) using the Fieldtrip toolbox [82].

## DWI acquisition and preprocessing (Experiment 1)

DWI data were collected from the same cohort using a 3.0 T MRI system (Simens Magnetom Prisma) with a 20-channel head coil with following parameters: repetition time (TR) = 4,000 ms, echo time (TE) = 79 ms, field of view (FOV) = 192 × 192 mm$^2$, voxel size = 1.5 × 1.5 × 1.5 mm$^3$, diffusion-weighted gradient directions = 64, b-values = 1,000 s/mm$^2$ and 2,000 s/mm$^2$, b0 nonweighted images = 5.

For DWI data, MRtrix3 and FSL software [83,84] were employed to preprocess the raw data. Preprocessing steps included denoising, reducing the ringing artifacts, eddy current and motion correction, and bias field correction using the

N4 algorithm provided in Advanced Normalization Tools. Gradient directions were also corrected after eddy current and motion correction.

## Music amplitude modulation spectrum (Experiment 1)

To derive the amplitude envelope of the musical stimuli in Experiment 1, we constructed 64 logarithmically spaced cochlear bands between 50 and 4,000 Hz using a Gammatone filterbank [85]. The amplitude envelope of each band was then extracted from the acoustic waveform using the Hilbert transform and down-sampled to 100 Hz. For each musical piece, the amplitude envelopes of the 64 bands were averaged, and the modulation spectrum of each piece was calculated using the Fast Fourier Transform (FFT) with zero-padding of 8,000 points, followed by taking the absolute value of each frequency point. Next, we normalized the modulation spectrum of each piece by dividing it by the norm of its modulation spectrum. Lastly, we estimated the mean amplitude modulation spectrum across the 10 pieces for each condition and showed them in Fig 1C.

## Blink and neural synchronization (Experiment 1–4)

To access whether blink and neural signals could synchronize to musical structures, we used a frequency domain analysis. The analysis window for blink data varied depending on the task. In summary, we analyzed the time series from 1 s after stimulus onset to 1 s before stimulus offset in Experiments 1 and 2, and we selected the time series from 1 s after stimulus onset to the onset of the first target in Experiments 3 and 4. For neural signals, the data from 1 s after stimulus onset to 1 s before stimulus offset was analyzed in Experiment 1. The blink and neural data for each trial and participant were transformed into the frequency domain using a FFT with zero-padding of 6,000 points. This achieved a consistent spectral resolution of 0.0167 Hz across the conditions, enabling precise quantification of the amplitude modulation spectra at the beat rate. To account for differences in trial length, we conducted surrogate test in the spectral domain to derive a null distribution of the amplitude modulation spectra for each independent variable. We then z-scored the raw amplitude spectra with respect to the surrogate mean, effectively normalizing them to remove biases introduced by varying stimulus durations. Next, we averaged the amplitude spectra across the 10 pieces for each condition. To identify robust blink and neural tracking of musical structures, we identified the lower and upper bounds of the significant frequency ranges and extracted average spectral amplitudes within these frequency ranges for further statistical analyses. For the baseline in Experiment 2, we first averaged the data across three trials in the temporal domain and then applied the same procedure for conversion to the frequency domain.

## Mutual information (Experiment 1)

In the above frequency-domain analysis, we observed robust blink synchronization to musical sequences primarily centered around the beat rate. To estimate the statistical dependency between the neural response and the eye-tracking data, we focused on the EEG power at the beat rate and computed MI between it and the blink signal. First, we extracted the time-frequency distribution of the EEG data at the beat rate (1.416 Hz) using a Morlet wavelet with a sliding window length of 3 cycles, as implemented in Fieldtrip. The analysis window spanned from 3 s before the onset to 3 s after the offset of each musical piece, with a step size of 10 ms. Next, we converted the output, corresponding to the EEG power, to a decibel scale using a pre-stimulus baseline ranging from −1,500 to −500 ms. Then, MI between the blink signal and EEG power was calculated using Gaussian Copula Mutual Information [86] at different lags: the blink signal was shifted against the EEG power from −200 to 200 ms in 20-ms steps. Finally, we summed the MI values across lags and averaged them across 10 pieces for each version and repetition for further analysis.

## Temporal response function (Experiment 1)

What's the relationship between ocular activity and neural response in the time domain? To reveal the temporal dynamics of the neural response to blink activity, we employed the mTRF toolbox [87] to estimate the TRF of EEG power at the beat

rate, utilizing the blink onset vector as the input stimulus feature. The TRF, denoted as w, was estimated using the following equation:

$$w = (S^T S + \lambda I)^{-1} S^T R \tag{1}$$

Here, S is the lagged time series of the input stimulus feature (the blink onset vector obtained from the eye-tracking data), R is the continuous neural response at the beat rate, I is the identity matrix, and $\lambda$ is a constant ridge parameter.

The TRF for each musical piece and participant was calculated using the blink onset vector and EEG power from 1 s after stimulus onset to 1 s before stimulus offset. The time lags ranged from 2,500 ms before blink onset to 2,500 ms after, with $\lambda$ set to 0. Then we averaged the TRF weights across 10 pieces for each version and repetition for further permutation test (described in the following section), and extracted the TRF weights at the peak time points for all independent variables for further comparison.

### Fiber tractography (Experiment 1)

Fiber tractography based on high angular resolution diffusion imaging (HARDI) was performed using MRtrix3. Firstly, 3-tissue (white matter, gray matter, and cerebrospinal fluid) response functions were obtained by the command "dwi2response dhollander" [88]. Secondly, fiber orientation distributions (FOD) of each voxel were estimated using multi-shell multi-tissue constrained spherical deconvolution algorithm [89]. Thirdly, the whole-brain probabilistic tractography was performed to generate 10 million streamlines for each participant based on iFOD2 tracking algorithm (step size = 0.75, angle threshold = 45°, maximum tract length = 150 mm, FA cutoff = 0.1) [90]. Lastly, track filtering was performed to attain 1 million streamlines [91].

The target segments of the AF and SLF were defined according to a simple classification system [92]: the dSLF, namely the classical SLF II, connects the Geschwind's area and the dorsolateral frontal area; the ventral SLF (vSLF) corresponds to the SLF III or anterior AF connecting the Broca's area and the Geschwind's area; the pSLF corresponding to the SLF-tp or posterior AF connects the Geschwind's area and the Wernicke's area; the AF, alternatively known as the classical AF or long AF, connects the Broca's area and the Wernicke's area. Freesurfer's automatic anatomical parcellation (aparc2009) algorithm [93] was used to define a set of 148 cortical and subcortical regions of interest (ROIs) from each individual's anatomical image. We further divided the superior temporal gyrus into equational anterior and posterior portions, and divided the precentral gyrus into equational dorsal and ventral parts. Four ROIs (Geschwind's area: supramarginal gyrus and angular gyrus; dorsolateral frontal area: posterior superior frontal gyrus and middle frontal gyrus; Broca's area: opercular part of inferior frontal gyrus and inferior part of precentral gyrus; Wernicke's area: posterior superior temporal gyrus and middle temporal gyrus) were extracted from individual anatomical image parceled by Freesurfer for each hemisphere to dissect bilateral segments of AF/SLF. S6 Fig shows fiber tractographies of the AF and SLF in bilateral hemispheres from one participant.

We extracted three indexes to measure the microstructural properties of fibers: FA derived from DTI as well as NDI and ODI derived from the NODDI model [94]. The mean FA, NDI and ODI values of each segment were extracted for each participant. LI for each parameter was subsequently calculated using the following equation:

$$LI = (left - right)/(left + right) \tag{2}$$

### Behavioral measure (Experiment 3 and 4)

Accuracy and reaction time were recorded in Experiments 3 and 4. Detection accuracy was defined as the ratio of correctly detected targets within one beat duration to the total number of targets presented. Only correct trials were included in the reaction time analysis.

**Blink probability (Experiment 4)**

In this analysis, we derived blink time series within a window from −0.353 to 0.353 s in relative to the beat onset, determined by the musical tempo (85 beats per minute). Blink probability at each time point was defined as the average value across trials for each participant, quantifying the likelihood of blink occurrence within the defined time window. Results are illustrated in S5 Fig.

**Statistical analysis**

All statistical analyses were calculated using MATLAB and SPSS (IBM Corp., Armonk, N.Y., USA). Group statistical analyses were performed on the data of all participants. Prior to analysis, outliers were identified and removed using the mean detection method.

For behavioral data, statistical differences between conditions (Figs 1D, 6E, and S1) were evaluated using a paired $t$ test with a threshold of $p = 0.05$. For the ocular and neural activities, all statistical analyses (Figs 2B, 3B, 3E, 5D left, and S3B) were performed with repeated-measures ANOVA to examine the effects of reversal manipulation, stimulus repetition, stimulus type and tempo. To address the issue of multiple comparisons, the statistical significance level was set at a corrected $p < 0.05$ using the Bonferroni correction method. The Greenhouse-Geisser correction was used when the assumption of sphericity was violated. All statistical tests performed were two-tailed. Correlations between blink/neural responses and behavioral indexes (Figs 2C, 5D right, 6C and S3C) were estimated using Spearman's rank correlation coefficients or Pearson's correlation coefficients according to the data distribution. Correlations between blink synchronization and neural entrainment (Fig 2E), as well as between blink/neural synchronization and white matter microstructural properties (Fig 4), were analyzed using partial correlations controlling for the MET rhythm score, with multiple comparisons corrected by the Bonferroni correction method.

To identify the significant frequency ranges showing blink/neural tracking of auditory sequences (Figs 2A, 2D, 5C, 6B, and 6D), we completed a surrogate test. Blink/neural data underwent the same Fourier transform processing steps described above, yielding a complex number at each frequency point for each trial and participant. Then the data were phase shuffled in the frequency domain by multiplying the data with a complex number which has a magnitude of 1 and a randomly generated phase between 0 and $2\pi$. As for the frequency domain analysis, we averaged the amplitude spectra of surrogate data across trials for each condition. This procedure was repeated 5,000 times, creating a null distribution of 5,000 surrogate amplitude modulation spectra for each condition, from which the 99th percentile was chosen as the significant threshold. Finally, we z-scored the raw modulation spectra with respect to the mean of the surrogate distribution for each condition. For baseline data (Fig 5B), the surrogate test procedure was identical, except for data surrogation in the temporal domain.

To assess the statistical significance of the coupling between the blink signal and the beat-rate EEG response (Fig 3A), we conducted a permutation test. Specifically, the beat-rate EEG power for each participant and trial was circularly shifted by a random shift amount within the time series, and MI between the time-shifted data and blink signal was recalculated. This procedure was repeated 5,000 times to create a null distribution of surrogate MI values for each version and repetition, from which a p-value was obtained by counting how many surrogate MI values exceeding the empirical MI value.

To confirm that the rise in TRF weights before blink onset was attributable to pre-onset prediction for the beat-rate neural response to blink (Fig 3D), we used a permutation test. The blink raw data for each participant and each trial was circularly shifted by a random shift amount within the time series, and the TRF was estimated between the new blink signal and EEG power. We repeated this process 1,000 times and a threshold of a one-sided alpha level of 0.01 was derived for statistical significance.

## Supporting information

**S1 Text. Differences between original and reverse music versions.**
(DOCX)

**S1 Fig. Comparisons of dynamic complexity (A), sensory dissonance (B), pitch salience (C), chord change rate (D), average loudness (E), root mean square (RMS) (F), beats per minute (G) and danceability (H) between the original and reverse versions of musical pieces.** $* p < 0.05$; n.s., not significant.
(TIF)

**S2 Fig. Blink probability at different time interval for two versions.** The length of each musical beat is 0.706 s. The figure displays that the inter-blink interval is most often one to two beats, consistent with what we observe in Fig 2.
(TIF)

**S3 Fig. Neural entrainment to musical beats and notes. (A)** Neural response spectra for two versions consistent with Fig 2D. The colored boxes indicate the frequency ranges where the amplitude was above the threshold. **(B)** EEG amplitude at note and beat rates. Error bars denote 1 SEM across participants. Note-rate EEG response increased with repeated exposure of music. **(C)** Correlation between the musical ear test (MET) rhythm score and EEG amplitude at note and beat rates. The two EEG responses were both correlated with the MET rhythm score for the reverse version. Colored dots indicate individuals. $* p < 0.05$, $** p < 0.01$.
(TIF)

**S4 Fig. Multiway canonical correlation analysis (MCCA). (A)** Contrasting scalp topographies of the first MCCA component during the third presentation of the original condition (left) and eye-related components extracted using independent component analysis (ICA) (right), highlighting clear spatial dissociation between neural and ocular sources. **(B)** Variance of summary components (SCs) extracted from MCCA. The variance of the first SC explains a considerable amount of variance. **(C)** Amplitude spectra of first three MCCA components for the third presentation in the original condition. The spectrum of the first component shows amplitude peaks corresponding to the beat rate and note rate (the first harmonic of the beat rate), further supporting the first MCCA component, but not other components, contained the neural signals induced by beat and note structures in the music pieces. Similar results are shown for other repetitions and other reversal conditions.
(TIF)

**S5 Fig. Blink probability in relative to the beat onset in Experiment 4.** Blinks were more likely to occur between two musical beats. The vertical dashed line represents beat onset. $N = 32$ participants.
(TIF)

**S6 Fig. The tractography of arcuate fasciculus (AF) and superior longitudinal fasciculus (SLF) in bilateral hemispheres from a participant.** Dorsal, ventral and posterior segments of SLF are depicted in purple, blue and green. The AF is depicted in yellow.
(TIF)

**S1 Table. Comparisons of key and scale between the original and reverse versions of musical pieces.**
(DOCX)

**S2 Table. Partial correlations between blink/neural synchronization and microstructural/laterality indices.** Values are $r$ ($p$). $P$ was estimated by partial correlation adjusting for the musical rhythm ability. * Bonferroni-corrected $p < 0.05$.
(DOCX)

**S3 Table. The mean and standard deviation of the blink rate for each condition after preprocessing.** Values are mean $\pm$ SD (units: blinks/min). bpm, beats per minute.
(DOCX)

## Acknowledgments

We thank Lidongsheng Xing and Jingwen Wang for their assistance with data collection; Pauline Larrouy-Maestri and Hou Chen for their assistance in preparing music materials; Baishen Liang, Xiang Li, Lei Zhang, Baihan Lyu, Shan Hua, and Peiqing Jin for their help with data processing. We thank Robert Zatorre for his comments on a previous version of the manuscript.

## Author contributions

**Conceptualization:** Xiangbin Teng, Yi Du.

**Formal analysis:** Yiyang Wu.

**Funding acquisition:** Xiangbin Teng, Yi Du.

**Investigation:** Yiyang Wu.

**Methodology:** Yiyang Wu, Xiangbin Teng, Yi Du.

**Supervision:** Yi Du.

**Visualization:** Yiyang Wu.

**Writing – original draft:** Yiyang Wu.

**Writing – review & editing:** Xiangbin Teng, Yi Du.

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
