## [Editor Report · Decision Letter 0]

22 Jan 2025

Dear Dr Du,

Thank you for submitting your manuscript entitled "Eye blinks synchronize with musical beats during music listening" for consideration as a Research Article by PLOS Biology.

Your manuscript has now been evaluated by the PLOS Biology editorial staff as well as by an academic editor with relevant expertise and I am writing to let you know that we would like to send your submission out for external peer review.

Once your full submission is complete, your paper will undergo a series of checks in preparation for peer review. After your manuscript has passed the checks it will be sent out for review. To provide the metadata for your submission, please Login to Editorial Manager (https://www.editorialmanager.com/pbiology) within two working days, i.e. by Jan 24 2025 11:59PM.

Kind regards,

Christian

Christian Schnell, PhD

Senior Editor

PLOS Biology

cschnell@plos.org

---

## [Decision Letter · Decision Letter 1]

2 Apr 2025

Dear Dr Du,

Thank you for your patience while your manuscript "Eye blinks synchronize with musical beats during music listening" was peer-reviewed at PLOS Biology. It has now been evaluated by the PLOS Biology editors, an Academic Editor with relevant expertise, and by several independent reviewers.

In light of the reviews, which you will find at the end of this email, we would like to invite you to revise the work to thoroughly address the reviewers' reports.

As you will see below, the reviewers expressed uncertainty about several key issues of the experimental design (e.g. repeated subjects), and we think that these and their other concerns will need to be fully addressed in the revision.

Given the extent of revision needed, we cannot make a decision about publication until we have seen the revised manuscript and your response to the reviewers' comments. Your revised manuscript is likely to be sent for further evaluation by all or a subset of the reviewers.

**IMPORTANT - SUBMITTING YOUR REVISION**

*Re-submission Checklist*

*Published Peer Review*

*PLOS Data Policy*

*Blot and Gel Data Policy*

Sincerely,

Christian

Christian Schnell, PhD

Senior Editor

PLOS Biology

cschnell@plos.org

REVIEWS:

Reviewer #1: The present paper investigates oculomotor activity, in particular eye blinks, while participants listen to music or sequences of tones - either attentively listening or performing a visual detection task. In addition, the authors measured EEG activity in one of the experiments and performed a link with acquired white matter data. The findings are discussed in relation to active sensing and dynamic attending frameworks.

The present study is timely and integrates in recent work using eye-movement measurements in listening tasks. While overall sound, I have the following concerns about the present manuscript, including its presentation. The nested, interlaced presentation of the 4 experiments is difficult to follow and is missing numerous information (see below). It would certainly be easier to follow - and also more complete - to change to a more classical presentation with methods/results/discussion for each of the experiments.

The integration of this work in previous recent research should be further developed, such as Jin et al. (2018). While briefly cited in the discussion, it should be further presented already in the introduction as this study shows how eye activity including blinks follow temporal regularity in speech and isochronous tone sequences. The here submitted study is thus not the first addressing this tracking possibility with blinks and this should be acknowledged from the start. It remains nevertheless an interesting, novel study using music together with other measures. The study presentation needs to further integrate in recent work on pupillometry, also showing the tracking of temporal regularity (e.g., Fink et al., 2018; 2024; Spiech et al., 2024) as do free eye movements (Schubert et al., 2025; Gehmacher et al., 2024). Discussion of these various measures together with the here acquired blink, EEG and anatomical data would provide a more complete picture for the reader and bring further our understanding of eye-behavior as a window into perception and cognition.

Material.

The authors used musical material in its original form, but also a reversed form. It is surprising that participants liked both versions equally well, with harmonic progressions and melodic/rhythmic features being altered. The reverse version reversed the "order of beats in each original piece". The authors should further analyze the created differences between original and reversed versions, notably in how far tonal and melodic structures and harmonic progressions were changed, as well as the tonal centers, pulse clarity, phrase structures etc. This seems relevant as tonal structures also guide temporal processing (e.g., what/when predictions, Jones & Boltz, 1989).

The authors adapted the original musical scores, for example eliminating rhythmic or acoustic cues for phrasal structures". The material is created and exported with MuseScore software. This software might include expressive performance cues (related to timing and intensity). The authors should confirm that they removed all these cues to create mechanical versions.

For each experiment, the auditory material should be made available to the readers as supplementary materials.

The reversed versions are designed as "unfamiliar control". However, when presented three times in a row, the item is not unfamiliar anymore. Can the authors further explain the rational of their design?

Line 532: "from 22.59 s to 59.30s" - why were musical pieces with this large range of durations used? Entrainment to the beat might differ between shorter and longer pieces. Different trial lengths also lead to different number of data points over the pieces, thus more or less stable "average amplitude spectra". Further, line 575: The rest period is of a duration of 3min. Why was this duration chosen, being different from the music durations?

Line 543: "we extract twenty-four melodic sequences" … consisted of eight chords". Please clarify the material description as chords contain at least two notes played simultaneously, thus differing from melodic sequences.

Line 564: The figure refers to repetitions of the same piece. Clarify "in a randomized order" here to avoid misunderstanding.

Line 571 "melody condition and rhythm condition" - does melody condition refer to the full original version (melody+harmony+rhythm) of Experiment 1? Please clarify labels.

Line 577: participants press the space bar as soon as possible. How does programming and realizing a motor response influence subsequent blinks?

Line 580: "participants were presented with two versions of the twenty-four melodic sequences" - What were these two versions? Original/reversed or rhythmic? Please clarify.

Line 588: "used as the original versions in Experiment 1" - please clarify.

Line 593: "each musical piece contained at least one target and up to three for each condition with the number of targets per condition was the same" - Please clarify which are the conditions and also how it is possible to have the same number of targets per condition in each piece when it varies from one to three.

Data analyses:

- The blink rate section (line 752ff) refers to 85 beats per minute. How were the other tempi analyzed? The window here is described as -0.353 to 0.353s.

- EEG data might be affected by blink occurrence and frequency. The authors comment on this (page 10) but should further explain this and provide the related data in the results section.

- Does task-related movement influence blink activity? If yes, how is this corrected?

Line 636: Add a reference justifying these limits.

Line 639: "data's mean" - is this the mean over trials in one participant or the group?

Line 650: MCCA: add some information presenting this method.

Line 681: "all independent variables" - please clarify which ones.

Line 744: Did the authors have hypotheses for laterality asymmetry?

Results:

The authors performed a median split which seems to require excluding 5 participants (Figure 4A). In light of Figure 4's distribution, one might wonder whether median split is appropriate here. Indeed, the median split approach with ANOVAs has been criticized (see, e.g., MacCallum, Zhang, Preacher, & Rucker, 2002, DOI: 10.1037//1082-989X.7.1.19). Including a continuous music background variable, e.g., as a continuous covariate in an AN(C)OVA, might be more appropriate.

Blink synchronization decreased with tempo, leading to non-significant findings for 120 bpm. How do the authors explain this? Might it be related to the fact that 500ms inter-beat-interval is too short regarding potential recovery periods?

Line 252: "Firstly ..." - which experiment is concerned here?

Participants:

Musical background is indicated with the Gold MSI output. Please explain to the readers what level of musical expertise is reflected by these numbers, and complete this information by the more usual description of years of musical training/instruction.

Discussion

Line 408: "motor cortical oscillations … actively predicts impending blink activation" - the wording must be turned down or completed with additional analyses.

Line 410: "the brain continuously updates its predictions" - also here, complete with the appropriate analyses over time.

Line 460: Discuss differences in materials and tasks in these studies to better understand previous findings and the present findings

Other points:

Line 375: "we then replicated this blink synchronization phenomenon across a broader range of tempi (66-120 bpm)…." VERSUS line 427: "we did not find significant blink synchronization at a tempo of 120 beats" - Please clarify results and their discussion.

Line 289: indicate the interval between beats in msec for readers unfamiliar with bpm.

Line 293: "Similar to Experiment 1, …….. detecting a timbre deviant" - please clarify if this task is new for Experiment 2 or clarify this task for Experiment 1.

Reviewer #2: This study investigates the association between the spontaneous eye-blinks and musical beat using eye-tracking, EEG recordings, white matter structural imaging, and behavioral analysis. The results showed synchrony between blink and the beat of music in original and reverse music versions. The analysis in the EEG recordings showed a relation between blink timing and neural beat tracking and differences were observed in the microstructure of the left arcuate fasciculus between low and high synchronizers. Finally, the authors established an association between blink synchronization and pitch deviant detection.

This is a large study combining eye-tracking, EEG recordings, white matter structural imaging, and behavioral analysis in order to quantify how spontaneous eye blinks synchronize with musical beats. The largest weakness of the papers lies in the fact that different subjects were used in the different experiments and that there is a lack of integration of the methodologies within a conceptual framework that integrates the observations into a hypothesis of the role of eye-blinking in beat perception. The study is descriptive in nature and does not establish a possible mechanism between audiomotor and eyeblinks for music beat perception and synchronization.

Minor comments.

Nothing is said in the introduction about the interaction between visuomotor rhythmic behavior, especially using rhythmic moving stimuli and its possible link with eye blinks.

On Figure 2A, it is difficult to see which repetition of the original and reverse conditions are above the threshold. In Figure 2B is not evident whether the blink amplitude is above random, especially for the third repetition, and the question is why.

There are inconsistencies in the effect of the condition (original and reverse) and the repetition across figures 2 and 3 that are not easily reconcilable.

The number of subjects for low and high blink synchronizers is small and the lack of differences in the two groups with their MET rhythm scores emphasizes the lack of consistency between blink behavior and rhythmic abilities in the subjects. It is not evident whether the authors used cluster correction analysis on their FA, NDI and ODI. It is difficult to associate the changes in the orientation dispersion index (ODI) of the left arcuate fasciculus in the two groups of subjects with the circuit that controls eye blinking.

Reviewer #3: The authors present 4 experiments in which they establish and replicate that blinks synchronize spontaneously to the beat structure of music, mediated by neural synchronization to the music. Blinks synchronize mainly to temporal rather than melodic properties of music, and seem to guide attention towards auditory but not visual targets. The authors further propose that individual differences in this blink-to-beat synchronization are explained by structural properties of the left arcuate fascicle. These are exciting new findings in the field of active sensing.

The study is timely, well conducted and very clearly presented. I receive it after the first revision, and obvious questions like whether the link between neural and blink synchronization is driven by spurious blink artifacts in the EEG signal have been ruled out. The perhaps weakest point in my view (although still interesting to be reported) is the trial to link blink synchronization to the microstructure of the AF. This is because low and high synchronizers do not seem to differ specifically in blink synchronization at the beat rate (S4 Fig.), but more globally. So, is it justified to call them "synchronizers" in the first place? Second, it is not clear whether this analysis was Bonferroni corrected (3 measures x 4 tracts x 2 hemispheres), but probably it was? Please clarify. Overall, I believe the study is well done and needs to be presented to the community. A few minor specifications may help readers to better understand or replicate the findings, as listed below, sorted by topic.

Interpretation suggestion:

- Line 448: The authors argue that the lack of blink synchronization in Experiment 4 is because of the task irrelevance of music. Have you considered that, alternatively, participants may have withheld or more voluntarily controlled blinks (e.g., blinked less) due to the visual task? Would a tactile task, i.e., a task that diverts attention from the auditory stream but isn't visual, be an alternative to dig deeper into the reasons for the lack of effects?

Methods specification:

- Line 250: "First, we excluded 3 outliers" — Outliers regarding which measure? Probably blink synchronization strength, but perhaps good to spell out.

- Line 254: "The blink amplitude spectra for the two groups showed a significant difference around the beat frequency (S4 Fig)." — If I understood correctly, the significant beat synchronization window was between 1.416-1.433 Hz (line 129). Why do the two groups show differences in synchronization across a much broader range, from 0.7 to 2.5 Hz? How specific is the median split to blink synchronization?

- Line 590: What duration did the red circle have?

- Line 637: Was blink rate calculated based on blink onset-to-onset intervals?

- Line 693: "a sliding window length of 3" — Please specify the unit. Seconds? Please also specify the frequency range and duration of the Morlet wavelet.

- Line 738: "parceled by Freesurfer" — Which atlas was used and which parcels in that atlas (please provide the names of the parcels)?

- Line 743: "mean FA, NDI and ODI values of each segment were extracted for each participant" — Maybe I overlooked it, but how were segments (tractography maps) thresholded?

- Line 752: It took me some time to understand that "blink rate" doesn't actually refer to a rate in Hz (as used in line 637), but something like a "blink probability" or "distribution around the beat". Perhaps this could be clarified.

- S5 Figure and line 468-470: "Blinks were more likely to occur around the beat onset" — Well, actually it looks as if most blinks occurred 300 ms before/after the beat, i.e., exactly between two beats. Is this really compatible with the interpretation of the post-blink boost effect?

- Generally, 1,000 permutations are rather at the lower end of permutation testing. 5k or 10k permutations would be better.

Typos detected on the way:

- Line 101: "while EEG and eye-tracking recordings simultaneously" — something is missing in this sentence.

- Line 400: Please correct "clarity" to "clarify".

- Line 409: Please correct "synchronizes" to "synchronize".

- Line 461: Please correct "other" to "others".

- Line 495: Please correct "found to implicated" to "found to be implicated".

- Line 570: Please correct "allowed" to "were allowed".

- Line 594: Please correct "with the number of targets per condition was the same" to "... being the same" or "with the same number of targets per condition".

- Line 638: Do you mean "Blinks with durations exceeding 3 SD" rather than "Blinks with mean durations exceeding 3 SD"?

- Line 734: Please correct "super marginal" to "supramarginal".

- Figure 1E: Does B and N on the x-axes of the power plots refer to beat-rate and note-rate? Please specify in the figure legend.

- Figure 1E: Why 90 trials? Weren't these 60 trials (10 pieces x 2 versions x 3 repetitions) in Experiment 1?

- Please check the references - formatting is not always consistent.

---

## [Decision Letter · Decision Letter 2]

22 Jul 2025

Dear Dr Du,

Thank you for your patience while we considered your revised manuscript "Eye blinks synchronize with musical beats during music listening" for consideration as a Research Article at PLOS Biology. Your revised study has now been evaluated by the PLOS Biology editors, the Academic Editor and one the original reviewers.

In light of the reviews, which you will find at the end of this email, we are pleased to offer you the opportunity to address the remaining points from Reviewer 1 in a revision that we anticipate should not take you very long. We will then assess your revised manuscript and your response to the reviewers' comments with our Academic Editor aiming to avoid further rounds of peer-review, although we might need to consult with the reviewers, depending on the nature of the revisions.

**IMPORTANT - SUBMITTING YOUR REVISION**

*Resubmission Checklist*

*Published Peer Review*

*PLOS Data Policy*

*Blot and Gel Data Policy*

Sincerely,

Christian

Christian Schnell, PhD

Senior Editor

PLOS Biology

cschnell@plos.org

REVIEWS:

Reviewer #1: The revised manuscript is improved and clarified. I nevertheless have the following comments that remain, but that the authors could certainly easily respond to in a revised version.

Material:

1) In their response "To further confirm the absence of expressive variation,..", the main point is not addressed as the comment here was about expressive features in the mechanical condition, thus still having somewhat expressive basic features that the software proposed. The authors response indicates that both original and reverse conditions do not differ in expressive variations. Please clarify.

2) The rational for the 'unfamiliar control' is still not clear. First, the authors explain that reverse versions still follow tonal harmonic structures, while then these are presented as unfamiliar in their progression.

3) Line 212 refers to the "unfamiliar harmonic progressions" requiring higher musical ability to track beat structures - but earlier in the manuscript, the authors claimed that the reversing did not influence perception.

4) The material construction requires some further explanations. Line 347 indicates the "new tone sequences maintaining the temporal structures of the original musical pieces but lacking pitch information or melodic cues", the figure suggests some adaptations, notably that two different tones were used (voice and bass line) and multiple tones were replaced by one.

5) - Line 660 (Exp3): Were the deviants always included in the same two pieces which "are interspersed within the block" ? if yes, the authors need to acknowledge that participants might learn where the deviant occurs (in the same two pieces) and just listen attentively to those but not all the others. this would considerably change the interpretation of the involved processes - and needs to be discussed in the manuscript.

6) line 722: how many on and off-beat targets occurred on average and indicate also SD and min/max

7) The varying duration of the different excerpts remains surprising. The analyses control for the length differences used by using zero-padding, however, this does not address the possibility that longer pieces allow for different beat entrainment strengths or processes. The authors could regroup the pieces as a function of duration and/or add duration as a predictor or a covariate.

Analyses/Results

Lines 697-699: Please add your rational of 6000 Data points for the FFT to share with the reader.

Please indicate how many blinks were actually performed during each of the musical pieces, that is the analyses are based on N blinks per trial (specified for each piece with its indicated duration)?

Line 350 "while their eye movements were recorded" ? only blinks were considered?

Results/Discussion etc.

- The lateralisation of the results should be further explained and integrated in the literature - why left- and not right-lateralized or bilateral? Furthermore, line 325 indicates that "better blink and neural synchronisation may be associated with lower neurone density in the left pSLF and AF". The question comes up whether it is about left-lateralisation or rather about the ratio of the involved structures across the two hemispheres and the "lower" indicates a benefit of enhanced asymmetry, thus requiring the inclusion of the right hemisphere to distinguish between these options?

- Line 443: The authors conclude that "blink synchronisation is a spontaneous behaviour induced by active engagement in music listening, rather than a response to passive auditory stimulus". However, the authors need also to consider the possibility that the visual task itself interferes with the eye-movements necessary for the task - and it is not related to passive versus active listening. - this would need to be acknowledged throughout the manuscript. The authors integrate it around line 555, but need to temper it elsewhere in the manuscript.

- The result with faster off-beat responses in the visual modality contradicts previous findings with better visual task performance for on-beat positions (e.g., Ecoffier et al., 2010; Trost et al., 2014; Bolger et al., 2013*) - the authors discuss potential differences with experimental paradigms (lines 555 etc), but it is not clear why these differences should actually matter and/or which underlying mechanisms involved are disturbed etc. Please clarify.

- Line 582: the observation that eye blinks are more likely to occur *between* two beats contradicts the claims throughout the manuscript that eye blinks "synchronise" to the beat.

Other points:

Wording is imprecise in several places, for example line 294 "modulated by cognitive factors, such as repetition and reversal manipulations" These later two refer to implementations and manipulations of the experimenter rather than cognitive factors or processes of the participants.

Idem line 393, it seems a stretch to refer to a deviant detection task as 'cognitive performance" - perceptual task?

Line 29: how is "strength" here defined?

Line 515: lower neuron density - left or right hemisphere?

*

https://pubmed.ncbi.nlm.nih.gov/20451167/

https://pubmed.ncbi.nlm.nih.gov/25224999/

https://pubmed.ncbi.nlm.nih.gov/23357092/

---

## [Decision Letter · Decision Letter 3]

18 Sep 2025

Dear Dr Du,

Thank you for your patience while we considered your revised manuscript "Eye blinks synchronize with musical beats during music listening" for publication as a Research Article at PLOS Biology. This revised version of your manuscript has been evaluated by the PLOS Biology editors, the Academic Editor and one of the original reviewers.

Based on the reviews and on our Academic Editor's assessment of your revision, we are likely to accept this manuscript for publication, provided you satisfactorily address the remaining points raised by the reviewer. Please also make sure to address the following data and other policy-related requests:

* Please add the links to the funding agencies in the Financial Disclosure statement in the manuscript details.

* DATA POLICY:

Regardless of the method selected, please ensure that you provide the individual numerical values that underlie the summary data displayed in the following figure panels as they are essential for readers to assess your analysis and to reproduce it: 1D, 3E, 6E and S1 (all panels).

* CODE POLICY

* Please integrate the text from the supplementary information and the references into the main manuscript file. We have no word count limitations.

We expect to receive your revised manuscript within two weeks.

*Published Peer Review History*

*Press*

Sincerely,

Christian

Christian Schnell, PhD

Senior Editor

cschnell@plos.org

PLOS Biology

Reviewer remarks:

Reviewer #1: I thank the authors for their thorough responses to my comments and suggestions. I just have the following points to be clarified in a revision:

- Line 448: yes, "task relevance", but please clarify that in the present case it is the visual task modality itself that might interfere with eye-movements necessary for the task and not related to passive/active listening.

- I understand the authors' analyses in the frequency-domain and that the focus is about the periodicity of blinking relative to the beat rather than blink count, thus leading to the FFT analyses. However, the report about how many blinks were performed per trial (or across trials, taking in consideration the duration) is still relevant as it provides the reader with the information about how many sample points are used to calculate the periodicity. Ideally, 1Hz corresponds to approximately 60 blinks, but you should tell readers the real numbers (how many blinks) that are used to calculate your periodicity. This could be done in a supplementary table listing both calculated periodicity Hz and number of sample points used to get this estimation.

---

## [Editor Report · Decision Letter 4]

8 Oct 2025

Dear Yi,

Thank you for the submission of your revised Research Article "Eye blinks synchronize with musical beats during music listening" for publication in PLOS Biology. On behalf of my colleagues and the Academic Editor, Mathew Diamond, I am pleased to say that we can in principle accept your manuscript for publication, provided you address any remaining formatting and reporting issues. These will be detailed in an email you should receive within 2-3 business days from our colleagues in the journal operations team; no action is required from you until then. Please note that we will not be able to formally accept your manuscript and schedule it for publication until you have completed any requested changes.

While you attend to those requests, please move the references from the SI file to the main reference list. References in the SI file are not picked up by the databases and the authors would therefore not get any recognition for their work.

PRESS

Sincerely, 

Christian

Christian Schnell, PhD

Senior Editor

PLOS Biology

cschnell@plos.org